# 3D evolution of Saharan dust transport towards Europe based on a 9-year EARLINET-optimized CALIPSO dataset

Eleni Marinou[1,2], Vassilis Amiridis[1], Ioannis Binietoglou[1,3], Athanasios Tsikerdekis[5], Stavros Solomos[1], Emannouil Proestakis[1,4], Dimitra Konsta[1], Nikolaos Papagiannopoulos[6], Alexandra Tsekeri[1], Georgia Vlastou[8], Prodromos Zanis[5], Dimitrios Balis[2], Ulla Wandinger[7], and Albert Ansmann[7]

[1]IAASARS, National Observatory of Athens, Athens, 15236, Greece
[2]Department of Physics, Aristotle University of Thessaloniki, Thessaloniki, 54124, Greece
[3]National Institute of R&D for Optoelectronics, Magurele, Romania
[4]Laboratory of Atmospheric Physics, Department of Physics, University of Patras, 26500, Greece
[5]School of Geology, Aristotle University of Thessaloniki, Thessaloniki, 54124, Greece
[6]Consiglio Nazionale delle Ricerche, Istituto di Metodologie per l'Analisi Ambientale (CNR-IMAA), Tito Scalo (PZ), Italy
[7]Leibniz Institute for Tropospheric Research, Leipzig, 04318, Germany
[8]Department of Physics, National and Kapodistrian University of Athens, Athens, Greece

*Correspondence to: Eleni Marinou (elmarinou@noa.gr)*

**Abstract.** In this study we use a new dust product developed using CALIPSO observations and EARLINET measurements and methods to provide a 3D multiyear analysis on the evolution of Saharan dust over North Africa and Europe. The product uses CALIPSO L2 backscatter product corrected with a depolarization-based method to separate pure dust in external aerosol mixtures and aSaharan dust lidar ratio based on long-term EARLINET measurements. The methodology is applied on a nine-year CALIPSO dataset (2007-2015) and the results are analyzed here to reveal for the first time the 3D dust evolution and the seasonal patterns of dust over its transportation paths from the Sahara towards the Mediterranean and Continental Europe. During spring, the spatial distribution of dust shows a uniform pattern over the Sahara desert. The dust transport over the Mediterranean Sea results in mean Dust Optical Depth (DOD) values up to 0.1. During summer, the dust activity is mostly shifted to the western part of the desert where mean DOD near the source is up to 0.6. Elevated dust plumes with mean extinction values between 10 - 75 $Mm^{-1}$ are observed throughout the year at various heights between 2 - 6$km$, extending up to latitudes of 40° N. Dust advection is identified even at latitudes of about 60° N, but this is due to rare events of episodic nature. Dust plumes of high DOD are also observed above Balkans during the winter period and above North-West Europe during autumn at heights between 2 - 4 $km$, reaching mean extinction values up to 50 $Mm^{-1}$. The dataset is considered unique with respect to its potential applications, including the evaluation of dust transport models and the estimation of cloud condensation and ice nuclei concentration profiles (CCN/IN). Finally, the product can be used to study dust dynamics during transportation, since it is capable of revealing even fine dynamical features such as the particle uplifting and deposition on European mountainous ridges such as the Alps and Carpathian Mountains.

## 1 Introduction

Mineral dust is ubiquitous in the atmosphere and one of the main contributors to the global aerosol load (Zender et al., 2004; Textor et al., 2006), with almost half of the global dust emissions generated in Africa (Huneuus et al., 2011). This has large consequences for air quality downwind (Viana et al., 2002; Gobbi et al., 2007), for the radiative budget due to scattering,
absorption, and emission of solar and terrestrial radiation (Balkanski et al., 2007), as well as for the cloud formation and lifetime (e.g., DeMott et al., 2003; Levin et al., 2005; Koren et al., 2010). These effects depend strongly on the vertical distribution of dust. For example, dust particles will have a stronger impact on shortwave radiation absorption when they are located above bright clouds (Yorks et al., 2009; Winker et al., 2013). Moreover, dust atmospheric lifetime is much longer in the free troposphere than in the planetary boundary layer, andupon entering the free troposphere, dust particles can be
transported across vast areas, altering the geographic pattern of their impacts (Prospero and Lamb, 2003; Levin et al., 2007; Ridley et al., 2012). Finally, the dust vertical distribution is crucial for dust-cloud interactions (e.g., Mamouri and Ansmann, 2016; Nickovic et al. 2016). Therefore, observing, monitoring and quantifying atmospheric dust burden and especially its vertical distribution is an important step towards understanding the climatic role of dust (IPCC, 2013, WG1, chapters 5, 7 and 9).

Lidar is the most prominent tool for aerosol profiling and has largely contributed to our knowledge of the vertical distribution of the dust optical properties (e.g., Liu et al. 2002; Ansmann et al., 2003; Balis et al., 2003; Papayannis et al., 2008; Mona et al., 2012; Granados-Muñoz et al. 2016; Bovchaliuk et al. 2016). Polarization lidar observations greatly expand the capabilities for dust detection, as non-spherical dust particles have a distinct signature on the particle depolarization ratio (e.g., Liu et al., 2008; Tesche et al., 2009). In Europe, the European Aerosol Research Lidar Network (EARLINET; Pappalardo et al., 2014)
operates advanced lidar systems employing depolarization techniques that have been invaluable for dust research. Sophisticated methodologies developed in EARLINET allow the complete characterization of different aerosol types including dust (e.g., Papayannis et al., 2008), as well as the dust contribution to the total aerosol load (Tesche et al., 2009).

The Cloud-Aerosol Lidar and Infrared Pathfinder Satellite (CALIPSO) mission equipped with the Cloud-Aerosol Lidar with Orthogonal Polarization (CALIOP) instrument has been delivering aerosol and cloud profiles across the globe for more than
ten years (Winker et al., 2009). This dataset offers the possibility to characterize the three-dimensional spatial distribution of aerosol as well as its temporal variation. CALIPSO is established as an accurate and robust mean for mineral dust identification from space (Liu et al., 2008; Omar et al., 2009). The application of EARLINET methodologies on CALIPSO observations can improve the observations for mineral dust research, as already suggested and applied in Amiridis et al. (2013). Specifically, this study retrieves the extinction of pure dust with high accuracy from CALIPSO, applying the depolarization-based
separation method introduced by Tesche et al. (2009), coupled with a regionally uniform climatological LR (lidar ratio) for calculating dust extinction. The latter value comes from long-term EARLINET measurements (Wandinger et al., 2010; Baars et al., 2016). It has been shown that the EARLINET-optimized CALIPSO dust product is in better agreement with Aerosol Robotic Network (AERONET) collocated measurements over Sahara and Europe and with Moderate Resolution Imaging

Spectroradiometer (MODIS) measurements over the Mediterranean for collocated cells with low cloudiness (Amiridis et al., 2013). This product is considered as the first accurate dust retrieval from space, since dust discrimination methods applied on passive sensors are based on the separation of the fine from coarse particle mode (e.g., Kaufman et al., 2005), delivering mostly biased DODs over the oceans due to the contamination of the coarse mode by sea-salt particles (Su et al., 2013). Another

advantage of the EARLINET-optimized CALIPSO dust product is its capability to provide accurate dust retrievals over all surface types, since the Cloud-Aerosol Lidar with Orthogonal Polarization (CALIOP) uses its own light source, overcoming the surface reflectance limitations of passive sensors (e.g., Hsu et al. 2004; Sayer et al., 2012).

Many studies have used satellite observations to derive dust properties over the Mediterranean during the last 15 years. Most of them focus on the horizontal distribution of dust using passive remote sensing techniques. Antoine and Nobileau, (2006)

used SeaWIFS (Sea-Viewing Wide Field-of-View Sensor) observations to study the seasonal evolution and variability of dust aerosols over the broader Mediterranean Sea during the period 1998-2004. Alpert and Ganor (2001) and Israelevich et al. (2002) used the Total Ozone Mapping Spectrometer (TOMS) Aerosol Index (AI) product in order to study the concentration of dust over Middle East and the dust sources of Northern Africa, respectively. The MODIS instrument, onboard both Terra and Aqua satellites has been extensively used in studies of airborne mineral dust over the Mediterranean basin. Barnaba and

Gobbi (2004) analysed one-year (2001) MODIS/Terra AOD at 550 nm observations and reported on the spatial distribution and seasonal variability of aerosols, including dust, over the Southern Europe, with a focus over the Mediterranean region. Papayannis et al. (2005) used MODIS/Terra data synergistically with lidar measurements and dust model simulations and investigated the vertical distribution of aerosols during dust outbreaks over Greece. Kosmopoulos et al. (2008) and Papadimas et al. (2008) used MODIS/Terra and MODIS/Aqua to investigate the seasonal and interannual variability of AOD at 550 nm

over Athens (Greece) and over the broader Mediterranean Sea, respectively. Marey et al. (2011) analysed ten-years of MODIS data synergistically with MISR and OMI and they produced a monthly climatology of aerosols over a domain covering the Nile Delta and northeast Africa. With respect to CALIPSO, the 3D distribution of dust and its optical properties have been studied for specific cases (e.g., Amiridis et al., 2009; Mamouri et al., 2009; Marey et al., 2011; de Meij et al., 2012; Nabat et al., 2012, 2013; Mamouri and Ansmann, 2015). Moreover, Winker et al. (2013) provided a 3D global aerosol climatology

from five-year CALIPSO data, along with the global distribution of mineral dust, derived using the ratio of columnar dust AOD to total AOD. Other studies offering a global view of desert dust using CALIPSO are provided in Liu et al. (2008a; 2008b), Adams et al. (2012), Yang et al. (2012), Tsamalis et al. (2013), Huang et al. (2015a; 2015b) and Gkikas et al. (2016). In particular, the studies of Liu et al. (2008b), Yang et al. (2012) and Tsamalis et al. (2013) examined the transatlantic Saharan dust transport focusing on the optical properties of dust, the influence of nearby clouds and the vertical distribution of the

Saharan Air Layer (SAL), respectively. Huang et al. (2015a) assessed the inferred most probable heights of global dust and introduced a separation method (Huang et al., 2015b) of anthropogenic dust (produced by human activities on disturbed soils) and free-tropospheric dust using CALIPSO and MODIS products. Liu et al. (2008a), Adams et al. (2012) and Gkikas et al. (2016) used CALIPSO observations in order to demonstrate the vertical structure of dust globally and/or above the Mediterranean. All the aforementioned studies are based on standard CALIPSO products with known limitations in accurately

typing and quantifying the optical properties of pure dust (Wandinger et al., 2010; Tesche et al., 2013). The EARLINET-optimized dust CALIPSO product presented herein, was used in Georgoulias et al. (2016a) to apply aerosol typing on MODIS and derive an aerosol climatology over the Eastern Mediterranean.

To our knowledge, this is the first time that a 3D pure-dust dataset is statistically analyzed over the area of North Africa and Europe in order to provide not only the horizontal but also the vertical patterns of Saharan dust intrusion in the Mediterranean. The study domain is from 20° to 60° N and from 20° W to 30° E. More specifically, we investigate the 3D inter-seasonal variation and intensity of dust transport patterns along with the inter-annual variations of DOD above this region. The paper is organized as follows: in Sect. 2 the CALIPSO lidar data are briefly introduced and the pure dust retrieval scheme is described in detail. In Sect. 3, the main findings are presented and discussed: Initially, the inter-seasonal variation and intensity of dust transport patterns (e.g. DOD, dust layer heights) are presented (Sect. 3.1-3.3), and the representative extinction coefficient values inside the dust plumes are derived (Sect. 3.4). In Sect. 3.5, the inter-annual variation of dust is examined while our summary and concluding remarks are given in Sect. 4.

## 2 Data and methodology

### 2.1. CALIPSO product

CALIOP, flying on-board the joint NASA/CNES CALIPSO satellite, delivers global aerosol and cloud profiles since June 2006 (Winker et al., 2009). CALIOP measures aerosol backscatter profiles at 532 nm and 1064 nm, including parallel and perpendicular polarized components at 532 nm, at high horizontal and vertical resolution. The data are processed to Level 2 (L2) products, providing aerosol and cloud backscatter and extinction coefficients at 532 nm and 1064 nm, as well as the linear particle depolarization ratio at 532 nm (Winker et al., 2009). First, the processing algorithm separates the atmospheric scene in distinct atmospheric layers (i.e. aerosol, cloud, surface returns; Vaughan et al., 2009). Then, for each aerosol layer the algorithm determines an aerosol subtype (i.e. dust, polluted dust, clean continental, polluted continental, marine, and smoke) based on a combination of information, such as the surface type, the layer integrated attenuated backscatter, the depolarization ratio at 532 nm and the aerosol layer height (Omar et al., 2009). The inferred subtype is used to derive the appropriate lidar ratio, a crucial input for the subsequent aerosol extinction retrieval (Young and Vaughan, 2009). Burton et al. (2013) showed an 80% successful detection of dust from CALIPSO, upon comparison to underflights with the HSRL system of NASA. This score is considered very high for aerosol typing purposes and is attributed to the depolarization measurement capability of the CALIOP sensor. Finally, the L2 products are aggregated to a gridded monthly-mean Level 3 (L3) product, providing mean profiles of extinction at 532 nm and mean AOD at a 2° × 5° spatial grid resolution (Winker et al, 2013). The most recent version of the L3 product (Version 3), released in October 2015, includes the correction of the AOD of cloudy scenes, the improved averaging of individual types as proposed by Amiridis et al. (2013) and Liu et al. (2008a), and corrections of signal artifacts responsible for high and low biases  as also observed in Papagiannopoulos et al. (2016).

## 2.2. EARLINET-optimized CALIPSO product

In this study, we make use of the EARLINET-optimized pure dust extinction product, monthly-averaged at a horizontal resolution of 1° × 1°, based on the methodology described in Amiridis et al. (2013). This product is a prominent outcome from the EARLINET-ESA collaboration for the LIVAS database (LIdar climatology of Vertical Aerosol Structure for space-based lidar simulation studies; Amiridis et al., 2015). Unlike the original CALIPSO L3 product of 2° × 5° resolution, the 1 degree resolution of LIVAS has been proved very useful in supporting studies of the same spatial resolution, specifically for the retrievals from passive satellite sensors and model evaluation studies (e.g., Popp et al. 2016, Georgoulias et al., 2016, Tsikerdekis et al., 2017). In our methodology, the pure dust backscatter coefficient ($\beta_d$) is decoupled from the total aerosol backscatter ($\beta_p$) based on depolarization measurements ($\delta_p$), assuming a particle depolarization ratio value for pure dust ($\delta_d$) equal to 0.31 (Tesche et al., 2009). Typical dust $\delta_d$ values measured with lidars in field campaigns around the globe are generally consistent with this value, showing little variation independently of the source region, (e.g., Sakai et al., 2000; Liu et al., 2008b; Freudenthaler et al., 2009; Groß et al., 2011; Burton et al., 2012; Burton et al., 2013; Groß et al., 2013; Groß et al., 2015; Illingworth et al., 2015). During SAMUM 1 and 2 campaigns Saharan dust $\delta_{nd}$ values varied between 0.27 and 0.35 at 532 nm (Ansmann et al., 2011), introducing 4% error in our calculations for the dust separated backscatter values. Using this separation technique, we avoid relying on the polluted dust and dust aerosol types used in CALIPSO, and thus, eliminate possible misclassifications found in CALIPSO L2 product (Burton et al., 2013). A final correction is related to the particle linear depolarization ratio, which is recalculated from L2 perpendicular and total backscatter profiles, to improve the accuracy compared to the original CALIPSO L2-Version 3 product, affected from a known bug (Tesche et al., 2013; Amiridis et al., 2013).

The quality control procedures and filtering criteria applied in the dataset are summarized in Table 1. In brief, CALIPSO L3 version 3 screening procedure is followed (Winker et al., 2013; CALIPSO L3-V3, 2015), and additional filters are incorporated to ensure the use of only cloud-free profiles. The additional methodology is as follows:

a) We remove all profiles with cloud features anywhere in the column.

b) We remove all profiles which fulfil the L3 CALIPSO "CAD score" or "Cirrus fringes" filters (see also Table 1).

The pure dust extinction coefficient is computed using a lidar ratio of 55 sr instead of 40 sr used in the CALIPSO product (Omar et al., 2009; Lopes et al., 2013). This value is representative of dust over Europe, mainly originating from Northwest Africa, as measured in coordinated CALIPSO/EARLINET measurements (Pappalardo et al., 2010; Wanginder et al., 2011), and is in excellent agreement with recent studies of dust measurements both near the source (Tesche et al., 2009; Veselovskii et al., 2016) and during long range transport (Preißler et al., 2011; Kanitz et al., 2013; Gross et al., 2015; Baars et al., 2016; Papagiannopoulos et al., 2016). The individual backscatter coefficient profiles at 532 nm are aggregated at a horizontal spatial resolution of 1° × 1° and a vertical resolution of 60 m from -0.5 $km$ to 20.2 $km$ and 180 $m$ from 20.2 $km$ to 30.1 $km$. Height is referenced to above sea level (a.s.l.) altitudes.

## 2.3. Climatological vs Conditional dust product

In this study, we calculate two separate dust products, the climatological and the conditional:

The climatological dust product is based on Amiridis et al. (2013), with a value of $0 \ km^{-1}$ assigned to the non-dust aerosol types when averaging within a cell. This product, hereinafter, is referred as Climatological Dust Extinction (Clim-DE) and the corresponding AOD as Dust AOD (DOD), and is presented and discussed in Sect. 3.1-3.3. As already discussed in the introduction, this product has been evaluated against AERONET data and is in very good agreement with collocated measurements over Sahara and Europe (Amiridis et al., 2013). The averaging methodology has been adapted by the L3-V3 CALIPSO product.

The conditional dust product is derived from averaging the CALIPSO dust extinction coefficients where dust is present, ignoring non-dust observations in the area. In particular, the clear air and non-dust aerosol types detected in the cell are ignored (set as NaN values when averaging). This product is referred to as Conditional Dust Extinction coefficient product (Con-DE) and is presented and discussed in Sect. 3.4.

The two products can be used for different applications: For example, Clim-DE is representative of the dust contribution to the total aerosol load and can be valuable in climatological studies. Moreover, the near-surface DOD provided by this product helps to estimate the natural aerosol contribution in the total aerosol load close to the surface for air-quality applications. The Con-DE product on the other hand, provides a measure of the intensity of the dust plumes.

## 2.4 Dust product uncertainties

The sources of uncertainties for the pure-dust product are discussed in this section. CALIOP is able to detect aerosol layers with $AOD > 0.005$ and $\beta > 0.25 \ Mm^{-1} \ sr^{-1}$ (Winker et al. 2009). The uncertainty estimation of particulate backscatter, extinction and AOD retrievals reported in the CALIPSO Level 2, Version 3 Data Release, are based on the simplified assumption that all the uncertainties are random, uncorrelated and produced no biases (Young, 2010). More specifically, ignoring multiple scattering, the errors in the layer optical depth calculations typically arise from three main sources: (a) signal-to-noise ratio within a layer, (b) calibration accuracy, and (c) the accuracy of the lidar ratio used for the extinction retrieval. The lidar ratio uncertainty is the dominant contributor to the total uncertainties, and the relative error in the layer optical depth is always at least as large as the relative error in the lidar ratio of the layer, and grows as the solution propagates through the layer (CALIPSO L2-V3, 2010). In our dataset the typical uncertainties in the CALIPSO Level 2 version 3 product are between 30% and 100% for the AOD, between 30% and 160% for the aerosol backscatter and extinction coefficient and $> 100\%$ for the particle depolarization ratio.

Several studies report that CALIPSO underestimates the columnar AOD due to undetected aerosol in the free atmosphere. For instance, Rogers et al. (2014) report a $\sim 0.02$ AOD CALIPSO underestimation, when compared to collocated airborne HSRL measurements over the North American and Caribbean regions at night. In their data, the dust layers were primarily non-

opaque with extinction less than $1 \, km^{-1}$ so there were negligible multiple scattering effects. The aforementioned detection limits and uncertainties of CALIPSO products are propagated to the dust product presented here.

As already described, the EARLINET-optimized CALIPSO dust product is derived using the depolarization-based separation method, coupled with the selection of a uniform climatological LR value. These steps introduce uncertainties in the pure dust product. In particular, the uncertainty in the selection of the representative LR ($55 \pm 11$) is 20% for the study area (e.g. Wandinger et al. 2010; Baars et al. 2016 and references within). This uncertainty in LR is less than half of the uncertainty of the generic LR in CALIPSO version 3 product ($40 \pm 20$ for dust layers and $55 \pm 22$ for polluted dust layers). As already addressed in several studies (e.g. Wandinger et al. 2010; Schuster et al. 2012; Amiridis et al. 2013), CALIPSO V3 dust extinction coefficient and AOD values are about 30% lower than those obtained from collocated ground-based Raman lidar retrievals due to the low LR used in the CALIPSO aerosol retrievals. Amiridis et al. (2013) applied the EARLINET LR for the pure dust CALIPSO cases above North Africa and Europe, and compared with synchronous and collocated AERONET measurements. The results showed an absolute bias on the AOD of the order of - 0.03, improving on the statistically significant biases of the order of - 0.10 reported in the literature for the original CALIPSO product. The bias of - 0.03 is similar to the low bias of CALIPSO's column AOD due to undetected aerosol layers. In Kim et al. (2017), they found a global mean undetected layer AOD of $0.0031 \pm 0.052$ by comparing 2 year of CALIPSO (L1-V4) and MODIS AODs.

Regarding the error induced from the application of the dust separation method, this might be due to the selection of the particle depolarization ratio of dust and the other aerosol types (marine, anthropogenic or smoke). Tesche et al. (2009; 2011) and Ansmann et al. (2012) estimated that the uncertainty in dust related backscatter coefficients is 15-20% in well-detected dessert dust layers and 20-30% in less pronounce aerosol layers. Moreover, we have calculated that the uncertainty of the dust occurrences presented in Sec. 3.1 ("% Dust / Used Overpasses"), might be up to 8% in latitudes away from the sources, induced from the error in the selection of the $\delta_{nd}$ value (0.03±0.04). Finally, an uncertainty induced in the dust product presented in this work, originates from the CALIPSO subtype selection algorithm. In this version of our product, both dust and polluted dust observations are considered polluted dust, and the pure dust component is separated using the dust separation method. The other aerosol layers, which are characterised as clean marine (CM), smoke (S), polluted continental (PC) or clean continental (CC) are considered to be cases clear of dust and are not tested for a dust component. This introduces negligible error in our analysis and is expected to induce a negative bias in the parameter "% Dust / Used Overpasses" less than 8%, mainly in areas above sea. In general, Clim-DE and Cond-DE products, the uncertainty of the dust extinction values close to the surface and at high latitudes is < 54%. At high altitudes and for latitudes up to 45°N, the uncertainty of the values is < 20%. Nevertheless, the standard deviation of the climatological products, coming from the natural variability of the dust events, may exceed to a large extent the uncertainty of the retrieval, reaching values as high as 100% and 200%.

In the latest release of CALIPSO Level 2 version 4 product (CALIPSO L2-V4, 2016), based on CALIPSO team announcement, the accuracy of the original CALIPSO product is increased and the uncertainty is reduced. This version is based on a revised calibration approach which leads to an increase in the total attenuated backscatter coefficients by ~3% overall as compared to

the version 3 values (CALIPSO L1-V4, 2016). Several bugs are fixed and a major overhaul of the aerosol subtyping algorithms along with revisions on the lidar ratio selections is applied.

## 2.5 Additional Satellite and Model dataset

The 6[th] version level-3 MODIS/Terra is a 1°x1° gridded aerosol dataset that is acquired from the NASA Giovanni system (https://giovanni.sci.gsfc.nasa.gov/giovanni/). In the current study the MODIS combined dataset, that takes into account the dark target (dark surface) and deep blue (bright surface) measurements, of aerosol optical depth was used for the period 2007 to 2015 (Sayer et al., 2014). Over the Mediterranean MODIS/Terra v6 was evaluated against 23 AERONET stations and was proven to score better that its predecessor MODIS/Terra v5 (Georgoulias et al., 2016).

The MACC global dataset is a reanalysis product based on the Integrated Forecast System (IFS) of the European Centre for Medium-Range Weather Forecast (ECMWF) coupled with the chemistry transport model MOZART-3 (Kinnison et al. 2007). The horizontal resolution of the model is 80km and it uses 60 vertical levels from the surface up to 0.1hPa. MACC was used in a numerous gas phase and particulate matter studies (Innes et al., 2013; Katragkou et., 2014; Eskes et al., 2015; Flemming et al., 2015; Cuevas et al., 2015; Georgoulias et al., 2017). The dust optical depth data used in this study covers the period 2007-2012 and all MACC data are open to the public (http://apps.ecmwf.int/datasets/data/macc-reanalysis/levtype=sfc/). RegCM4 is an open source area-limited, sigma-p vertical coordinated regional climate model (Giorgi et al., 2012) based on the hydrostatic core of the PSU/NCAR Mesoscale Model (MM5; Grell et al., 1994). The simulation used in the current study is a part of a previous research where the dust optical depth of the model was evaluated against the dust climatological product of this work, after it was fully spatially and temporally collocated with the exact flyby of CALIPSO (Tsikerdekis et al., 2017). The simulation covers the period 2007 to 2014 with a horizontal resolution of 50km and 18 vertical sigma-p levels.

## 3 Results and discussion

In Sect. 3.1 - 3.4, we examine the inter-seasonal variation and intensity of dust transport patterns, from 2007 to 2015, for the domain 20° W to 30° E and 20° N to 60° N. In Sect. 3.1 we provide the average climatological state of the seasonal dust distribution at a spatial resolution of 1° × 1°. In Sect. 3.2 we give information on dust layer heights. In Sect. 3.3, we illustrate the mean climatological vertical structure of dust reaching Europe. To achieve that, the area of study is separated into five longitudinal zones with a step of 10°. In Sect. 3.4, we illustrate the vertical intensity of the dust plumes, using again longitudinal zone maps. Finally, in Sect. 3.5, we examine the inter-annual variation of dust.

## 3.1 Horizontal dust distribution

In this section, we provide the average climatological state of the seasonal horizontal dust distribution derived from CALIPSO dust product at a spatial resolution of 1° × 1° for the domain of North Africa and Europe. The seasonal grouping used in this study is as follows: from January to March (JFM), from April to June (AMJ), from July to September (JAS) and from October

to December (OND). In our study region, March and October are considered transition months for Saharan dust advection (e.g., Ganor, E., 1994; Guirado et al. 2014). This grouping is based on the dominant patterns revealed from the maps of monthly mean DODs (not shown). More specific, the decision is based on the observation that the events during February–March and October–November, although rarer, are usually more intense than those of the other months. This is further supported from a

ten year (2001 – 2011) analysis of African dust outbreak $PM_{10}$ observations over the Mediterranean basin (Pey et al., 2013). Figure 1 shows the geographical distribution of dust occurrences (Figs. 1a, c, e, g) and the corresponding mean DOD values for each season (Figs. 1b, d, f, h).In order to provide a more quantitative representation of the dataset, the domain is aggregated in six areas over the study region. The main results and statistical parameters are provided in Table 2. In particular, the information provided is: the mean and standard deviation of the DOD, the maximum values along with the 95% percentile in

parenthesis, the layer's centre of mass and top height along with their standard deviations (these parameters are discussed in the next section), the percentage of the observations with DOD greater than zero in the cloud-free observations (with 100% as unity), the percentage of the cloud-free occurrences in the total observations provided by the CALIPSO product (with 100% as unity) and each domain's geographic extent.

Table 2 shows the impact of cloud contamination in our dataset. During AMJ, JAS and OND, more than 80% of the total

observations are cloud-free above North Africa. Above Central-East Mediterranean (C-E Med.), more than 80% of the total observations are cloud-free and above Central West Mediterranean (C-W Med.) approximately 60% - 80% of the total observations are cloud-free. With increasing latitude, the cloud-free sampling is reduced to percentages of ~ 40% - 60% in latitudes greater than 45° N. During JFM, cloudy conditions restrict our dataset in the greatest extent. During the same period, the cloud-free cases used represent ~ 80% of the total observations above North Africa, approximately 60 - 70% of the total

observations above the Mediterranean and ~ 30% in the domain between 45° N - 60° N. In the areas (and seasons) where clouds do not dominate (e.g. 70% clear-sky conditions), our cloud-free product is considered representative of the dust distribution. In areas where cloudy skies dominate (e.g. 30% clear-sky conditions), the clear-sky CALIPSO profiles cannot be considered as representative of all meteorological conditions, so the results should be used with caution.

Based on Fig. 1 and Table 2, the overall percentages of dust occurrences and mean DOD values are greater during summer

and spring months. During autumn and winter the emission and transport of dust towards Europe is suppressed due to the more effective removal processes and due to the atmospheric dynamics favouring the transport of dust towards the Atlantic (e.g. Israelevich et al., 2002; Schepanski et al., 2009). More specific, during JFM (Figs. 1a, b) limited dust activity is observed almost uniformly over the Sahara desert. The DOD remains roughly over the entire study domain below 0.13 with 75% of the observations having DODs < 0.17, 95% of the observation having DODs < 0.5 and extreme values with DODs ~2. The dust

occurrences decrease with latitude and the presence of dust is approximately 70 % over Africa and the Mediterranean region and decreases to lower than 50 % over northern Europe. The most affected area during these months is eastern Mediterranean. The cyclone formation over the central Mediterranean, which is affected by mid-latitude depressions generated either in the Atlantic Ocean or in north-western Europe (e.g., Trigo et al., 1999; Maheras et al., 2001), results in the transportation of dust from the Libyan Desert towards the Balkans which leads to dust occurrences up to 70 % (Fig. 1a) along with mean seasonal

DOD of 0.1-0.2 (Fig. 1b). In the domains between 10° E - 30° E and 30° N - 40° N, 5% of the dust events are observed with DODs > 0.41, 1% with DODs >0.95 and extreme observations with DODs are up to 1.6. Similar mean values have been reported in the literature for this period, along with extreme events characterized by AOD values higher than 1 (Gerasopoulos et al., 2011). Moving northward, mean DOD tends decrease due to the increasing distance from the major dust sources and

5 also due to higher precipitation at the northern parts of the study region that efficiently removes dust from the atmosphere (e.g., Moulin et al., 1998; Marrioti et al., 2002).

During AMJ (Figs 1c, d) dust production occurring over the entire Saharan desert with mean DOD values of 0.26 ± 0.26 and occurrences of 86%, uniformly at latitudes between 20° N and 30° N. The activated dust sources are located in the broad "dust belt" and are usually associated with topographical lows in the arid regions and with the intermountain basins (Prospero et al.,

2002). The arrival of mid latitude extratropical cyclone systems from the Atlantic Ocean as well as cyclogenesis at the Gulf of Genoa and/or at northern African coast favours dust transport over central and eastern Mediterranean. Mean DOD over these areas reaches values of 0.12 ± 0.20 (Fig. 1d) and extreme observations observed with DODs up to 2.74. Dust is also present over central and northern Europe with occurrence percentages between 65 % and 53 % (Fig. 1c; Table 2), revealing that dust particles can be transported far away from their sources under favourable meteorological conditions.

During JAS (Figs. 1e, f), intense dust activity is largely shifted to the western part of the Sahara where dust occurrences are >90 % and mean DOD near the sources up to 0.6 (Fig. 1f). In the domain between 10° W - 00° and 20° N - 35° N, the mean DOD is 0.43, with 25% of the dust observations having DODs > 0.69, 5% >1.2 and the extreme DODs up to 3 (Table 2). The migration of the ITCZ (Intertropical Convergence Zone) towards higher latitudes and the dominance of trade wind patterns (easterlies) benefit the transportation of dust towards the Atlantic Ocean as seen also by the westward plumes in Figs. 1e and

1f. In the same period, increased dust occurrences (83 %) are also found over Western Mediterranean and South Italy. In the domain between 10° W - 00° and 35° N - 45° N, the mean DODs are 0.09 ±,0.14 with 5% of the dust observations having DODs >0.55 and extremes DODs up to 2.3.

During OND dust activity is significantly suppressed (Fig. 1g) except from the south-west desert areas close to the Sahel where mean DOD lies in the range 0.2-0.3 (Fig. 1h). In the domain between 10° W - 00° and 20° N - 30° N, the mean DODs are 0.43

± 0.39 and extremes DODs up to 3 (Table 2)).

In order to provide a more informative representation of the dust product presented here, we performed a comparison with MODIS AOD for the same period and the dust optical depth of the MACC reanalysis and a RegCM4 simulation for the period 2007-2012 and 2007-2014 respectively (Fig. 2). MODIS provides AOD for all natural and anthropogenic aerosol types. As a result the MODIS average value for the whole period and domain (0.267) is 281% almost three times, bigger than our product

(0.095). It is noted thought that the values between the two satellite products are very similar over the Sahara desert. On the contrary, the corresponding average dust optical depth values of MACC (0.100) and RegCM4 simulations (0.104) reproduce better our product, since only dust is considered, though our product is lower by 5% and 8.6% respectively. Dust optical depth is overestimated over Europe and Mediterranean by MACC and RegCM4 simulations in comparison to our product in all seasons and especially in the hot periods AMJ and JJA, but the reasons of these discrepancies have to be further studied.

### 3.2 Vertical dust distribution

CALIPSO offers the ability to assess the vertical distribution of dust from space. To facilitate the investigation of the vertical characteristics of dust, two parameters are introduced, the dust Top Height (TH) and the dust Centre of Mass height (CoM) (Mona et al., 2006; Mona et al., 2014; Binietoglou et al., 2015). TH is defined as the height corresponding to the altitude where the 98 % of the dust extinction lies below. CoM is estimated by the calculation of the extinction-weighted altitude given by the formula:

$$CoM = \frac{\int_{z_t}^{z_b} z a(z) dz}{\int_{z_t}^{z_b} a(z) dz},$$ (1)

where $z_b$ and $z_t$ are the base and top altitude of the dust feature respectively. α denotes the dust extinction coefficient at altitude z. The estimate of CoM provides information related to the altitude where the most part of the dust load is located. This parameter is considered ideal for comparisons with aerosol layer height retrievals from passive remote sensing (e.g., IASI, GOME-2A, Sentinel5P and the future Sentinel-4 and Sentinel-5 missions (Ingmann et al. 2012)), since these retrievals are sensitive to the location of the dust mass maximum within the layer (e.g., TROPOMI Aerosol Layer Height product; Sanders et al. 2015).

Figure 3 shows the spatial distribution of TH and CoM for the four seasons. In Table 2, the TH and CoM values (a.s.e.) are accompanied with their standard deviations providing an indication of the variability of the dust heights in the atmosphere of the study area. During JFM dust resides in general below 3 $km$ a.s.e. (above surface elevation) over land with CoM at about $1.3 \pm 1.6$ $km$ a.s.e. (Figs. 3a, b). Over the sea, several transport paths are discernible especially over eastern Mediterranean with dust tops traveling at $2.3 \pm 1.9$ $km$ a.s.e. During AMJ, TH and CoM are up to $4.2 \pm 1.7$ $km$ and around $2.4 \pm 1.1$ $km$ a.s.e. respectively over eastern parts of Sahara. Over the Mediterranean Sea and South Europe dust tops extend around 2-3.5 $km$ and CoM around 1-2 $km$ a.s.e., with Centre and East Mediterranean having the most elevated plumes (Figs. 3c, d). The latitudinal slope of CoM denotes the latitudinal transport of dust during AMJ from south to north. The highest TH values (>4.5 $km$) are found during the warm period (JAS) over north-western Africa and over the adjacent Atlantic Ocean region (Figs. 3e, f). This is most likely attributed to the intrusion of the lower tropospheric Atlantic monsoon, south of the ITCZ, and the development of MCS (mesoscale convective systems) that favour the elevation of dust at this area (Bou Karam et al., 2008). The dust height decreases towards the eastern part of the study region. In the interim, the dominance of the strong Saharan high enables the mobilization of dust from western part of Sahara towards western Mediterranean and Europe. This pattern leads to elevated dust at $3.0 \pm 1.7$ $km$ a.s.e. and CoM at $1.6 \pm 1.1$ $km$ a.s.e. over south European countries and Balkans. During OND the horizontal pattern is similar to JJA however with much lower heights (Figs. 3g, h).

In general, our results are in agreement with lidar-based studies which have been performed in several European sites. Papayannis et al. (2008) performed an exhaustive analysis on Saharan dust particles over Europe using EARLINET lidar profiles. They found that the dust layer center of mass extends from 3.0 to 3.8 km and the thickness ranges from 0.7 to 3.4 km. Specifically, Balis et al. (2012) calculated the mean base and top of dust layers in the eastern Mediterranean, Thessaloniki, to

be around $2.5 \pm 0.9\ km$ and $4.2 \pm 1.5\ km$, respectively. More recently, Mona et al. (2014) analyzed a long dataset of Saharan dust intrusions over Potenza, Italy, and found a mean layer centre of mass of $3.5 \pm 1.5\ km$.

## 3.3 Climatological dust cross sections

To further illustrate the vertical dynamics of dust reaching Europe, the area of study between 20° W and 30° E is separated into five longitudinal zones of 10° interval, covering latitudes from 20° to 60° N. The vertical structure of the averaged Climatological Dust Extinction coefficient (Clim-DE) for each of these five longitudinal zones (illustrated in Fig. 4 as latitude-height cross-sections) reveal several dust layers and strong seasonal variations. The two dashed lines are drawn such as to show how many dust observations are averaged for the extinction retrievals. The extinction values below the higher dashed line have been calculated by averaging a number of dust observations, greater than 18 (2 dust overpasses per season and year). The extinction values below the lower dashed line have been calculated by averaging a number of dust observations greater than 54 (2 dust overpasses per month and year). The median surface elevation is depicted with black colour (and is labeled as NaN) in the plots. In general dust is always ubiquitous at heights close to the surface throughout the year. The lower layers are representative of near source dust activity and boundary layer processes. The spring and summer peaks indicate the increased activity of Saharan dust sources (Moulin et al., 1998; Schepanski et al., 2007). More specific, for the area between 10° W and 20° W over the Atlantic extending from Africa to west of Spain and Ireland, the presence of elevated dust plumes is evident (Figs. 4a-d) mainly during summer and for latitudes up to 30°N. During JFM the plume is located below 2 $km$ height a.s.l. (above sea level), while from spring to autumn the plume reaches a height of 5 $km$ a.s.l. and yields high values of extinction coefficient (~75 $Mm^{-1}$) over Africa. Over the area from 0° to 10° W, extending from western Algeria, Morocco, Spain, and the British Isles we found Clim-DE values inside the Africa mixing layer greater than 60 $Mm^{-1}$ for all seasons. Maximum values of extinction are observed during summer months when dust is elevated up to 6 $km$ with Clim-DE values around 120 $\pm$ 140 $Mm^{-1}$ above N. Africa and mean values exceeding 200 $Mm^{-1}$ above the Algerian Desert (Fig. 4g). These findings are in good agreement with more than two years of AERONET observations in Tamanrasset site, a strategic site for dust research located in the heart of Sahara (Guirado et al., 2014). A steep decrease in extinction values is observed along the African coastline with values of 20 $Mm^{-1}$ above the southern part of the Iberian Peninsula (38°-42° N) where dust is trapped by the Pyrenees. The distinct decrease of extinction values across the African coastline is an indication that dust is always present inside the rather deep Saharan boundary layer while it is only occasionally transferred towards the Mediterranean when atmospheric dynamics favor this kind of flow. At higher latitudes, the CALIPSO dust extinction is drastically reduced but still observed in ranges of 1-2 $km$ a.s.l. and with mean Clim-DE values of 5 $Mm^{-1}$. Moving eastwards (0°-10° E) elevated dust is trapped topographically by the Alps (47°-52° N) with values >10 $Mm^{-1}$. As the dust-laden air-masses approach the mountains, they decelerate and thus the dust concentration increases (Israelevich et al., 2012). Maximum values of extinction (>50 $Mm^{-1}$) are observed over northern Africa during summer (Fig. 4k). Close to the Algerian sources, south of the Atlas Mountains (~30° N) extinction coefficient is greater than 200 $Mm^{-1}$ close to the surface (Fig. 3k). The area south of Atlas Mountains (Fig. 4e,

f, g, h) is characterized by haboob activity (Knippertz et al., 2009; Solomos et al., 2012). These systems are generated from convective outflows and contribute to the interannual burden of dust at this area. As dust extends to higher latitudes (30°-40° N) Clim-DE decreases (<75 $Mm^{-1}$). Over the area between 10°-20° E (Figs. 4m-p), similar patterns are observed. This region includes the dust sources of Libya and central Sahara, central Mediterranean, the eastern Alps and part of North Europe. It is evident from this figure that dust extinction over central Mediterranean (35°-45° N) is around 25 $Mm^{-1}$ throughout the year. As in the previous western zonal section the same pattern over the Alps is encountered. Moving further eastwards maximum values of Clim-DE are found during spring. At the most eastern part of the study area (20°-30° E; Figs. 4q-t), dust is trapped by the Carpathian Mountains (45°-49°N) especially during winter, thus highlighting, once more, the role of topography. Significant dust presence is evident all over the zonal section (until 60° N) mostly attributable to elevated dust traveling along with the westerlies from western and central parts of Europe towards East. Above the Balkans and during JFM values of $29 \pm 65\ Mm^{-1}$ are observed in the first 1.5 $km$, and $10 \pm 30\ Mm^{-1}$ between $2.5 - 3.5\ km$. In AMJ and JAS respectively, means of $\sim 16 \pm 40\ Mm^{-1}$ and $\sim 9 \pm 20\ Mm^{-1}$ are observed in altitudes between 1.5 to 5 $km$. The values of Clim-DE are higher (>45 $Mm^{-1}$) over Africa during winter and spring, in relation with the ones observed during the other two seasons (<45 $Mm^{-1}$) and reach high altitudes (5-6 $km$ a.s.l.) during spring and summer. In summary, the obtained cross-sections for the five longitudinal zones indicate that higher extinction coefficient values are observed near the source and at low altitudes, where dust particles are efficiently deposited. Above NE Africa, the Clim-DE values are >45 $Mm^{-1}$ throughout the year in altitudes up to 2 $km$ a.s.l. during JFM and up to 4 km during AMJ and JJA. Moreover, the standard deviation of the means is around 130% at the altitudes up to 2 km and ~100% between $2 - 4$ km, at all seasons. Above West Africa, the extreme Clim-DE values observed during JAS in the altitudes up to 2 km are $113 \pm 131\ Mm^{-1}$. In C-E Mediterranean, dust is always present, with maximum extinctions during AMJ, reaching $27 \pm 54\ Mm^{-1}$ close to the surface and $\sim 18 \pm 30\ Mm^{-1}$ during JAS and OND. In C-W Mediterranean, the highest means of JAS are $\sim16 \pm 40\ Mm^{-1}$. For latitudes greater than 45° N, and during AMJ mean values of $8 \pm 27\ Mm^{-1}$ are $4 \pm 16\ Mm^{-1}$ are observed close to the surface above NE Europe and NW Europe respectively. The above results can be used to also estimate the airborne mass concentration of dust. The dust mass concentration can be obtained from the optical properties of dust with an uncertainty of 20-30% (Ansmann et al., 2012; Mamouri and Ansmann, 2014). For example, the Clim-DE values imply dust mass concentration >75 $\mu g\ m^{-3}$ above Africa throughout the year and >125 $\mu g\ m^{-3}$ above West Africa during JAS. In South Europe and Mediterranean, the corresponding values are >17 $\mu g\ m^{-3}$ in the first 2 $km$ a.s.l. and ~50 $\mu g\ m^{-3}$ close to the surface. For latitudes greater than 45° N, values around 8 $\mu g\ m^{-3}$ are the most common. The decreasing intensity with height and latitude found in the Clim-DE product is representative of the average dust distribution over the area. However, this behaviour is not representative of the distribution during dust episodes over Europe. This is because the extinction coefficient values presented in Fig. 4 for the Clim-DE product are produced by averaging partially and fully dominated dust cases. In order to describe the spatial patterns and the intensity of the dust plumes during episodes only, we introduce and discuss the Con-DE product in the next section.

## 3.4 Conditional dust cross sections

The two dashed lines in Fig. 5 correspond to the number of dust observations greater than 18 (2 dust overpasses per season and year) and greater than 54 (2 dust overpasses per month and year) for the lower and higher dashed line respectively. Con-DE values derived from less than 4 dust observations (dO) in each cell are masked with grey colour (and are labeled as <4dO) in the plots. The median surface elevation is depicted with black colour (same as in Fig. 4). Con-DE values are significantly different from the Clim-DE, as seen in Fig. 4. Although Con-DE has similar values to Clim-DE near the sources, where dust is always present, above the Atlantic and the Mediterranean Con-DE is characterized by significantly higher values. This is because the two products differ mostly over areas which are not dominated by dust. In the vertical cross sections of Fig. 5 the pattern of Con-DE shows two distinct dust features. For example over the longitudinal zone from 20° to 30° E during summer (Fig. 5o) two dust features are visible: one above North Africa extending from the surface to ~5 $km$ a.s.l. and another above the Mediterranean between 3 and 6 $km$ a.s.l. The two distinct layers are also identified in other regions and in other seasons (e.g. Fig 5a, l, p, s, t). These populations are linked to two different processes: the near surface dust at the southern parts of the study region represents fresh emissions from the dust sources, while the elevated plumes that extend northern until 40° N are due to the advection of dust, associated to the seasonality of the long-range transport paths (Lelieveld et al., 2002; Israelevich et al., 2012; Huneeus et al. 2016). This separation is enhanced as one moves from the west to the east sectors. At the western part of the domain (10°-20° W) the near surface and elevated dust originates probably from the same sources. Similar double layer patterns are found in all seasons and over all areas with various characteristics. For example during JAS at the region extending from 0° to 10° W (Fig. 5g) the generation of dust from the source region is much more intense than the transportation term, which is also evident. For the same period, in the area 0° to 10° E the transportation term above the Mediterranean between 3 and 6 $km$ height, originating from the intensive source regions, becomes much more important than the source term at the same cross section.

Looking in more detail the vertical cross sections of Fig. 5, we observe the rare but very intense elevated dust plumes during JFM (4a, e, i, m). During that period, dust is advected between 1.5–4 $km$ height a.s.l. with Con-DE values >45 $Mm^{-1}$ equivalent to dust mass concentrations >75$\mu g\ m^{-3}$. Furthermore, in Fig. 5q, the intensity of the JFM dust episodes above the Balkans is depicted. The Con-DE value in this domain is in the same range as the one in the other regions at the same period, but the dust plumes can be thicker, extending from the ground until 4 $km$ a.s.l. The trapping of Saharan dust from the mountainous ridges of Europe (located between 40°N – 50 °N, e.g. the Alps 45°N-48°N) is also evident by the Con-DE cross-sections(e.g. Fig. 5i, m). The deceleration of the transport air masses along the mountain ridges results to the accumulation of dust at the windward slopes. Dry deposition of dust at these areas result also in the formation of "brown snow" and albedo reduction, with profound climatological implications (e.g., Fujita, 2007; Shahgedanova et al., 2013). This phenomenon is more intense during JFM period due to the advection of dust at lower heights.

During AMJ (5b, f, j, n, r) and JAS (5c, g, k, o, s) the elevated dust above Mediterranean has Con-DE values of 35-50 $Mm^{-1}$ (58-83 $\mu g\ m^{-3}$), in heights between 2-6 $km$ and up to latitudes of 40° N. The transport of dust during AMJ is mostly due to

the eastward propagation of N.Africa – Mediterranean low pressure systems (Sharav cyclones). Dust is embedded in the cyclonic circulation and the penetration to latitudes higher than 40°N is limited. For latitudes 40° and 50° N, during the warm seasons (AMJ and JAS), the Con-DE values inside the transported dust plumes is between 20-40 $Mm^{-1}$ everywhere (33-67 $\mu g\ m^{-3}$). Rare events, characterized by relatively higher Con-DE (>35 $Mm^{-1}$and >58 $\mu g\ m^{-3}$), between 2-5 $km$ a.s.l., are observed over the British Isles and Germany during OND (Figs. 5h, l). These events, caused by the propagating low pressure systems over the East Atlantic, have been documented in detail from the EARLINET community reporting extinction coefficient values up to 200 $Mm^{-1}$ inside dust plumes (Ansmann et al., 2003; Müller et al., 2003). In the vertical cross section of Fig. 5 it is evident that dust reaches the upper levels of the troposphere (>8 $km$ a.s.l.) with Con-DE values of ~10 $Mm^{-1}$ in all longitudinal zones and during all seasons. Dust occurrence is very low, close to zero for heights greater than 8 $km$ a.s.l. during spring and summer and for heights greater than 6 $km$ a.s.l. during autumn and winter. ). A quantitative representation of the Clim-DE and Con-DE products is provided in Table 3. In this, regional statistics on the two products, along with their standard deviation are provided for three altitudinal ranges ($0 – 2, 2 – 4$ and $4 – 6\ km$ a.s.l.).

### 3.5 Interannual variability of dust

In this section we use the CALIPSO derived monthly mean DOD values, for the total-column and for five individual layers (0.18–0.5, 0.5–1, 1–2, 2–4, 4–8 $km$), to study their inter-annual variability during the 9 year period between 2007 and 2015. The selected layers are representative for both near surface and long-range transported dust plumes. The data are aggregated on a 10° x 10° cell over the study region. Using a first-order autoregressive linear regression model on the de-seasonalized monthly DOD values as described in Zanis et al. (2006), temporal trends of DOD were calculated. Figure 6 shows the geographical distribution of de-seasonalized trends ($year^{-1}$) for the columnar DOD (a) and for the five individual layers (b-f). Hatched filled grid cells depict the statistical significance trends with 99% confidence. A decrease ~ 0.001 $yr^{-1}$ (~ 4% $yr^{-1}$) is evident for the South European cells (0°-30° E, 40°–50° N) (with these values being > 95% statistically significant). Examination of the five vertical layers shows a similar decreasing pattern. The negative trends observed in the area (mainly above North Africa and Mediterranean), are additionally characterized by constant decrease throughout the layers, although the trends are not statistically significant. The small negative DOD trends (< 0.002 $yr^{-1}$corresponds to < 5% $yr^{-1}$) coming from this temporal limited dataset, are in good agreement with the global decrease of dust estimated from an 161-year time series of dust from 1851 to 2011, created by projecting wind field pattern onto surface winds from a historical reanalysis in Evan et al. (2016), and also with the global mean near-surface dust concentration decrease by 1.2% $yr^{-1}$ reported in Shao et al. (2013) paleoclimate research for the period 1984-2012 (even though Europe and North America are excluded from their trend analysis). In comparison with studies relevant to the time period considered in this work, the DOD decrease of 0.001 $yr^{-1}$ over the northern coast of Africa is in agreement with Floutsi et al. (2016), who based on 12 years of MODIS-Aqua observations (2002-2014) reported an average decrease of 0.003 $yr^{-1}$ for the coarse mode fraction of AOD over the broader Mediterranean Sea. Furthermore, over the same domain the decreasing trend of DOD coincides with the decrease of Saharan

desert dust episodes as reported by Gkikas et al. (2013). Regarding the AERONET stations over the domain of northern Africa and Europe, Yoon et al. (2012) reported on the trends of AOD at 440 nm along with the corresponding Ångström Exponents (440 and 870nm). The documented negative trends over the AERONET stations of Avignon (France), Dakar (Senegal) and Ispra (Italy) are in agreement with the negative DOD reported here, although with discrepancies in the magnitude, while trend

disagreements are observed over the AERONET station of Banizoumbou (Niger). The decreasing trends of DOD observed over the domain northern of Africa and Europe coincide with the generally documented downward AOD trends reported based on several satellite observations of MODIS/Aqua, MODIS/Terra, MISR and SeaWiFS (Pozzer et al., 2015; de Meij et al., 2012; Hsu et al., 2012; Georgoulias et al. 2016b). More particular, in the most recent study of Georgoulias et al. (2016b), using MODIS/Terra and MODIS/Agua observations, they reported negative statistically significant trends over Algeria, Egypt and

the Mediterranean and positive trends over Middle East. Overall, for the Mediterranean they reported an AOD trend of -0.0008 $yr^{-1}$ for the MODIS/Terra observations (2000 – 2015) and -0.0020 $yr^{-1}$ for the MODIS/Aqua observations (2002 – 2015), with the trends being statistical significant at the 95% confidence level in both cases. A possible increase is only found for west Sahara areas (10° - 0° W, 20°–30° N). However, the results for this cell are not considered statistical significant. In our study, we calculate the DOD trend along with the statistically significance of each trend for the period 2007-2015 (108 monthly

values). Nine years are considered a small period for a robust trend calculation and it would be interesting to repeat the same analysis in the future to extended aerosol record. The de-seasonalization process as well as the trend are describing the examined period only. Figure 7 shows the DOD internal variability of the 20 individual areas, as it is calculated from monthly mean DODs. Is evident from this figure that the DOD values in 2008 are relatively higher than the other years and in almost all the domains bellow 40°N. Similarly, relatively high values are observed in some of these areas for the year 2010. Since

these years are at the beginning of our study period, they have a significant contribution on the negative trends observed during the examined period.

## 4 Summary and conclusions

An optimized CALIPSO dust product was recently developed with a regional correction of the Saharan dust lidar ratio using EARLINET measurements (Amiridis et al., 2013). The same product has been used here to provide the three-dimensional dust

distribution and its transport pathways across northern Africa and Europe. The monthly climatology of African dust obtained from 2007 to 2015 allows the description of the spatio-temporal features of dust properties. The study of the mean state climatology shows strong seasonal shifts in dust source regions and transportation pathways. The seasonal cycle of the dust transport is well captured with the lowest values of DOD in winter and the highest values in spring and summer. During summer and autumn, dust aerosols are more confined to the source region, while during spring significant dust aerosols from

the Sahara are extended uniformly over North Sahara and are transported over the Mediterranean and Europe. Dust extinction coefficient, Centre of Mass and Top Height retrieved parameters are used to quantitatively understand the 3D evolution of dust and the seasonal differences in the vertical distribution which are evident as well. Over the source region of Sahara Desert,

dust CoM and the TH are higher during spring and summer and lowest during winter. Dust transport mechanisms are more efficient during summer when dust is often lifted up to $6\ km$, coinciding with the deepest dust layer. The appearance of localized regions of increased extinction coefficient values over mountains (the Alps, the Pyrenees, and the Carpathian Mountains) denotes the existence of aerosol transport routes that decelerate in front of the mountain ranges. Rare and intense events are observed above Balkans during winter period and above North-West Europe during autumn. The inter-annual analysis revealed that DOD trends during the study period are of the order of 0.001/year for the South European cells, showing constant decrease throughout the different layers.

The dust climatology presented here is of paramount importance in understanding the three-dimensional production and transport of Saharan dust which will contribute to better estimate the dust impacts on climate. The use of two products, the climatological and conditional in this study allowed us to conclude on both the dust contribution to the total aerosol load over our domain, but also on the intensity of the Saharan dust events recorded in the region. Future work includes: (i) the optimization of CALIPSO dust retrievals based on measured dust LR from ground-based lidars and particle depolarization ratio over extended regions of deserts in the Middle East and China, to obtain a robust global climatology of dust; (ii) the calculation of cloud condensation nuclei and ice nuclei concentrations from polarization lidar as suggested by Mamouri and Ansmann (2016), to provide a quantification of the climatic effect of dust on cloud formation.

## 5 Data availability

The CALIPSO data were obtained from the online archive of ICARE Data and Services center http://www.icare.univ-lille1.fr/archive (CALIPSO Science Team, 2015; ICARE Data Center, 2016). MODIS data are publicly available on the NASA Giovanni system (https://giovanni.sci.gsfc.nasa.gov/giovanni/). MACC data are publicly available on http://apps.ecmwf.int/datasets/data/macc-reanalysis/levtype=sfc/. The regional climate model ReGCM4 code is available at https://gforge.ictp.it/gf/project/regcm/frs/. RegCM4 simulation data used in this work are available upon request from Athanasios Tsikerdekis (tsike@geo.auth.gr). The LIVAS database is publicly available at http://lidar.space.noa.gr:8080/livas/. LIVAS EARLINET-optimized pure dust products are available upon request from Eleni Marinou (elmarinou@noa.gr) and Vasilis Amiridis (vamoir@noa.gr).

## Acknowledgements

The authors acknowledge support through the following projects and research programs:
- ESA-ESTEC project LIVAS (contract N°4000104106/11/NL/FF/fk)
- BEYOND under grant agreement no. 316210 of the European Union Seventh Framework Programme: FP7-REGPOT-2012-2013-1

- MarcoPolo under grant agreement n° 606953 from the European Union Seventh Framework Programme (FP7/2007-2013)

- ACTRIS under grant agreement no. 262254 of the European Union Seventh Framework Programme: FP7/2007-2013

- ACTRIS-2 under grant agreement no. 654109 from the European Union's Horizon 2020 research and innovation programme

- ITaRS under grant agreement no. 289923 of the European Union Seventh Framework Programme: FP7/2007-2013

- ECARS under grant agreement No 602014 from the European Union's Horizon 2020 Research and Innovation programme

- EPAN II and PEP under the national action "Bilateral, multilateral and regional R&T cooperations" (AEROVIS Sino-Greek project)

- A. G. Leventis Foundation scholarship

The authors acknowledge EARLINET for providing aerosol lidar profiles available under the World Data Center for Climate (WDCC) (The EARLINET publishing group 2000-2010, 2014 a, b, c, d, e). We thank the AERONET PIs and their staff for establishing and maintaining the AERONET sites used in this investigation. CALIPSO data were obtained from the ICARE Data Center (http://www.icare.univ-lille1.fr/). CALIPSO data were provided by NASA. We thank the ICARE Data and Services Center for providing access to the data used in this study and their computational center. We thank Jason Tackett for his support during the algorithm development for the production of Level 3 CALIPSO products.

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

**Tables and Figures**

**Figure 1:** Geographical distribution of the seasonal dust occurrences (a ,c ,e ,d) and the mean DOD values (b ,d ,f ,h) for the three-month averages: January–March (a ,b), April-June (c ,d), July–September (e ,f), and October–November (g ,h), and the domain between 20° W-30° E and 20°-60° N  for the period 2007-2015, measured with the CALIPSO climatological dust product.

**Figure 2:** Comparison of the seasonal spatial distribution of the optical depth as received by (first column) pure-dust CALIPSO DOD product, (second column) MODIS AOD product, (third column) MACC reanalysis DOD product, (fourth column) RegCM4 simulated DOD product.

**Figure 3:** Geographical distribution of the dust top height (a-d) and the center of mass (e-h) in $km$ a.s.e. measured with CALIPSO dust product for the three-month averages: January–March (a, e), April-June (b, f), July–September (c, g), and October–November (d, h), and the domain between 20° W-30° E and 20°-60° N for the period 2007-2015.

**Figure 4:** Geographical zonal distribution of the climatological dust extinction coefficient values ($Mm^{-1}$) measured by CALIPSO dust product for the regions 10° W to 20° W (a-d), 0° to 10° W (e-h), 0° to 10° E (i-l), 10° E to 20° E (m-p), and 20° E to 30° E (q-t) for the latitudinal regions from 10° N-60° N as illustrated by domain maps for the three-month averages: January–March (a, e, i, m, q), April-June (b, f, j, n, r), July–September (c, g, k, o, s), and October–November (d, h, l, p, t). The median surface elevation is depicted with black colour.

**Figure 5:** Geographical zonal distribution of the conditional dust extinction coefficient values ($Mm^{-1}$) measured by CALIPSO dust product for the regions 10° W to 20° W (a-d), 0° to 10° W (e-h), 0° to 10° E (i-l), 10° E to 20° E (m-p), and 20° E to 30° E (q-t) for the latitudinal regions from 10° N-60° N as illustrated by domain maps for the three-month averages: January–March (a, e, i, m, q), April-June (b, f, j, n, r), July–September (c, g, k, o, s), and October–November (d, h, l, p, t). The median terrain elevation is depicted with black colour.

**Figure 6:** Geographical distribution of the de-seasonalized trends ($yr^{-1}$) derived from monthly columnar DOD (a) and for five individual layers (b-f), for the period 2007-2015, aggregated over 10° x 10° grid cells. Hatched filled grid cells depict the statistical significance trends with 99% confidence.

**Figure 7:** Interannual variability of the DODs for the 10° x 10° grid cells depicted in Fig. 6, for the period 2007-2015.

**Table 1: Quality control procedures and filtering applied in CALIPSO data.**

1. Screen out all features that are not aerosols

2. Set all clear air profile measurements to 0.0 $km^{-1}$

3. Samples below opaque cloud and aerosol layers are removed

4. Clear-Sky Mode: Only measurements in which no clouds are in the column are considered

5. Large negative near-surface extinction filter: all level 2 aerosol extinction samples adjacent to the surface having a value less than -0.2 $km^{-1}$[1] are ignored

6. Samples where aerosol extinction uncertainty is less than 99.9 $km^{-1}$ are allowed

7. CAD score: Only features having cloud-aerosol discrimination (CAD) scores between -100 and -20 are used

8. Only features having extinction QC flag values of 0, 1, 16, or 18 are allowed

9. Cirrus Fringes: Misclassified cirrus in the upper troposphere, coming from CAD artifacts, are removed

10. Remove measurements which are contaminated by surface values: extinction values near the surface less than $-0.2\ km^{-1}$ are ignored

11. Undetected surface attached aerosol low bias filter (Changed between CALIPSO L3 version 1 and version 3): samples classified as "clear air" lying beneath the lowest quality screened aerosol layer whose base is below 250 m from the local surface are ignored

12. Negative signal anomaly mitigation strategy: all level 2 aerosol extinction coefficients within 60 meters of the planetary surface are excluded from level 3 calculations (new in L3 version 3)

13. All non-dust aerosol types detected in the cell are assigned with a value of 0.0 $km^{-1}$

Extra filters with more strict cloud screening:

14. All profiles having cloud features anywhere in the column are removed

15. All profiles which fulfil the L3 CALIPSO "CAD score" or "Cirrus fringes" filters are removed

**Table 2: Regional statistics on mean dust optical depth, max values, dust layer center of mass (CoM) and top height (TH) (a. s. e.), ratio of dust observations to cloud-free observations, ratio of cloud-free observations to total observations and domain boundaries.**

| | DOD Mean ± St.dev. | DOD Max Vals. (Perc. 95%) | CoM ± St.dev. | Top Height ± St.dev. | Nr Dst in Nr cl-free | Nr cl-free in Nr obs. | Domain |
|---|---|---|---|---|---|---|---|
| **NE Africa** | | | | | | | |
| JFM | 0.11 ± 0.17 | 2.19 (0.42) | 1.5 ± 1.2 | 2.6 ± 1.8 | 0.72 | 0.84 | [10E,30E] |
| AMJ | 0.26 ± 0.26 | 3.09 (0.73) | 2.4 ± 1.1 | 4.2 ± 1.7 | 0.86 | 0.86 | [20N,30N] |
| JAS | 0.18 ± 0.21 | 2.63 (0.56) | 2.3 ± 1.0 | 4.0 ± 1.4 | 0.84 | 0.93 | |
| OND | 0.11 ± 0.14 | 2.93 (0.34) | 1.9 ± 0.9 | 3.3 ± 1.4 | 0.81 | 0.93 | |
| **NW Africa** | | | | | | | |
| JFM | 0.13 ± 0.18 | 1.86 (0.47) | 1.5 ± 1.3 | 2.4 ± 1.8 | 0.67 | 0.82 | [10W,10E] |
| AMJ | 0.26 ± 0.26 | 2.31 (0.75) | 2.2 ± 1.2 | 3.8 ± 1.6 | 0.86 | 0.83 | [20N,35N] |
| JAS | 0.43 ± 0.39 | 3.03 (1.20) | 2.9 ± 1.0 | 5.1 ± 1.3 | 0.94 | 0.88 | |
| OND | 0.22 ± 0.26 | 2.59 (0.71) | 2.2 ± 1.0 | 3.9 ± 1.6 | 0.82 | 0.81 | |
| **C-E Med.** | | | | | | | |
| JFM | 0.09 ± 0.18 | 1.62 (0.41) | 1.3 ± 1.4 | 2.3 ± 1.9 | 0.69 | 0.70 | [10E,30E] |
| AMJ | 0.12 ± 0.20 | 2.74 (0.51) | 1.8 ± 1.5 | 3.2 ± 2.1 | 0.82 | 0.76 | [30N,45N] |
| JAS | 0.08 ± 0.12 | 1.80 (0.33) | 1.6 ± 1.1 | 3.0 ± 1.7 | 0.89 | 0.96 | |
| OND | 0.08 ± 0.11 | 1.55 (0.31) | 1.4 ± 1.1 | 2.7 ± 1.6 | 0.82 | 0.80 | |
| **C-W Med.** | | | | | | | |
| JFM | 0.03 ± 0.06 | 1.09 (0.11) | 1.3 ± 1.6 | 2.0 ± 1.9 | 0.49 | 0.57 | [10W,10E] |
| AMJ | 0.05 ± 0.10 | 1.35 (0.25) | 1.8 ± 1.6 | 2.9 ± 2.2 | 0.65 | 0.61 | [35N,45N] |
| JAS | 0.09 ± 0.14 | 2.33 (0.36) | 1.9 ± 1.2 | 3.3 ± 1.8 | 0.75 | 0.80 | |
| OND | 0.05 ± 0.09 | 1.62 (0.20) | 1.3 ± 1.2 | 2.3 ± 1.6 | 0.63 | 0.64 | |
| **NE Europe** | | | | | | | |
| JFM | 0.025 ± 0.055 | 0.97 (0.11) | 1.2 ± 1.4 | 1.7 ± 1.7 | 0.37 | 0.28 | [10E,30E] |
| AMJ | 0.033 ± 0.062 | 1.61 (0.12) | 1.6 ± 1.2 | 2.5 ± 1.6 | 0.61 | 0.47 | [45N,60N] |
| JAS | 0.032 ± 0.045 | 0.90 (0.11) | 1.6 ± 1.1 | 2.7 ± 1.4 | 0.60 | 0.58 | |
| OND | 0.023 ± 0.043 | 0.50 (0.09) | 1.2 ± 1.0 | 1.9 ± 1.4 | 0.49 | 0.43 | |
| **NW Europe** | | | | | | | |
| JFM | 0.015 ± 0.033 | 0.47 (0.06) | 1.2 ± 1.6 | 1.7 ± 1.7 | 0.36 | 0.36 | [10W,10E] |
| AMJ | 0.023 ± 0.037 | 0.73 (0.08) | 1.5 ± 1.6 | 2.2 ± 1.9 | 0.52 | 0.47 | [45N,60N] |
| JAS | 0.022 ± 0.042 | 0.93 (0.08) | 1.4 ± 1.5 | 2.1 ± 1.7 | 0.43 | 0.52 | |
| OND | 0.018 ± 0.035 | 0.57 (0.07) | 1.1 ± 1.2 | 1.7 ± 1.4 | 0.40 | 0.44 | |

**Table 3: Regional statistics on the dust extinction coefficient for altitudes between 0 to 2km, 2 to 4 km and 4 to 6 km (a. s. l.).**

| | 0 – 2 km | 2 – 4 km | 4 – 6 km | |
|---|---|---|---|---|
| | Clim-DE / Cond-DE / St. dev | Clim-DE / Cond-DE / St. dev | Clim-DE / Cond-DE / St. dev | Domain |
| **NE Africa** | | | | |
| JFM | 42 / 50 / 74 $Mm^{-1}$ | 7 / 43 / 20 $Mm^{-1}$ | 0 / 25 / 5 $Mm^{-1}$ | [10E,30E] |
| AMJ | 66 / 66 / 88 | 44 / 53 / 48 | 18 / 48 / 26 | [20N,30N] |
| JAS | 42 / 42 / 64 | 30 / 40 / 37 | 13 / 43 / 22 | |
| OND | 34 / 34 / 51 | 17 / 32 / 24 | 3 / 27 / 9 | |
| **NW Africa** | | | | |
| JFM | 46 / 60 / 80 $Mm^{-1}$ | 6 / 45 / 18 $Mm^{-1}$ | 0 / 29 / 5 $Mm^{-1}$ | [10W,10E] |
| AMJ | 73 / 73 / 90 | 41 / 59 / 49 | 13 / 51 / 25 | [20N,35N] |
| JAS | 113 / 113 / 131 | 83 / 83 / 71 | 43 / 50 / 40 | |
| OND | 59 / 59 / 86 | 35 / 48 / 43 | 10 / 36 / 19 | |
| **C-E Med.** | | | | |
| JFM | 22 / 44 / 55 $Mm^{-1}$ | 4 / 48 / 16 $Mm^{-1}$ | 0 / 31 / 5 $Mm^{-1}$ | [10E,30E] |
| AMJ | 27 / 35 / 54 | 17 / 52 / 34 | 5 / 42 / 15 | [30N,45N] |
| JAS | 18 / 18 / 28 | 13 / 33 / 22 | 4 / 37 / 12 | |
| OND | 19 / 23 / 32 | 10 / 35 / 19 | 2 / 27 / 7 | |
| **C-W Med.** | | | | |
| JFM | 5 / 24 / 33 $Mm^{-1}$ | 1 / 32 / 7 $Mm^{-1}$ | 0 / 21 / 2 $Mm^{-1}$ | [10W,10E] |
| AMJ | 10 / 23 / 38 | 6 / 35 / 19 | 1 / 31 / 8 | [35N,45N] |
| JAS | 16 / 22 / 40 | 13 / 33 / 23 | 5 / 38 / 14 | |
| OND | 10 / 22 / 33 | 4 / 29 / 14 | 0 / 29 / 4 | |
| **NE Europe** | | | | |
| JFM | 4 / 37 / 41 $Mm^{-1}$ | 0 / 29 / 5 $Mm^{-1}$ | 0 / 15 / 1 $Mm^{-1}$ | [10E,30E] |
| AMJ | 8 / 17 / 27 | 2 / 21 / 17 | 0 / 14 / 2 | [45N,60N] |
| JAS | 7 / 14 / 21 | 2 / 16 / 9 | 0 / 16 / 2 | |
| OND | 4 / 16 / 19 | 1 / 21 / 6 | 0 / 14 / 1 | |
| **NW Europe** | | | | |
| JFM | 1 / 16 / 16 $Mm^{-1}$ | 0 / 16 / 2 $Mm^{-1}$ | 0 / 15 / 1 $Mm^{-1}$ | [10W,10E] |
| AMJ | 4 / 16 / 16 | 1 / 21 / 11 | 0 / 14 / 2 | [45N,60N] |
| JAS | 3 / 15 / 15 | 1 / 22 / 7 | 0 / 18 / 2 | |
| OND | 2 / 16 / 15 | 0 / 23 / 4 | 0 / 13 / 0 | |

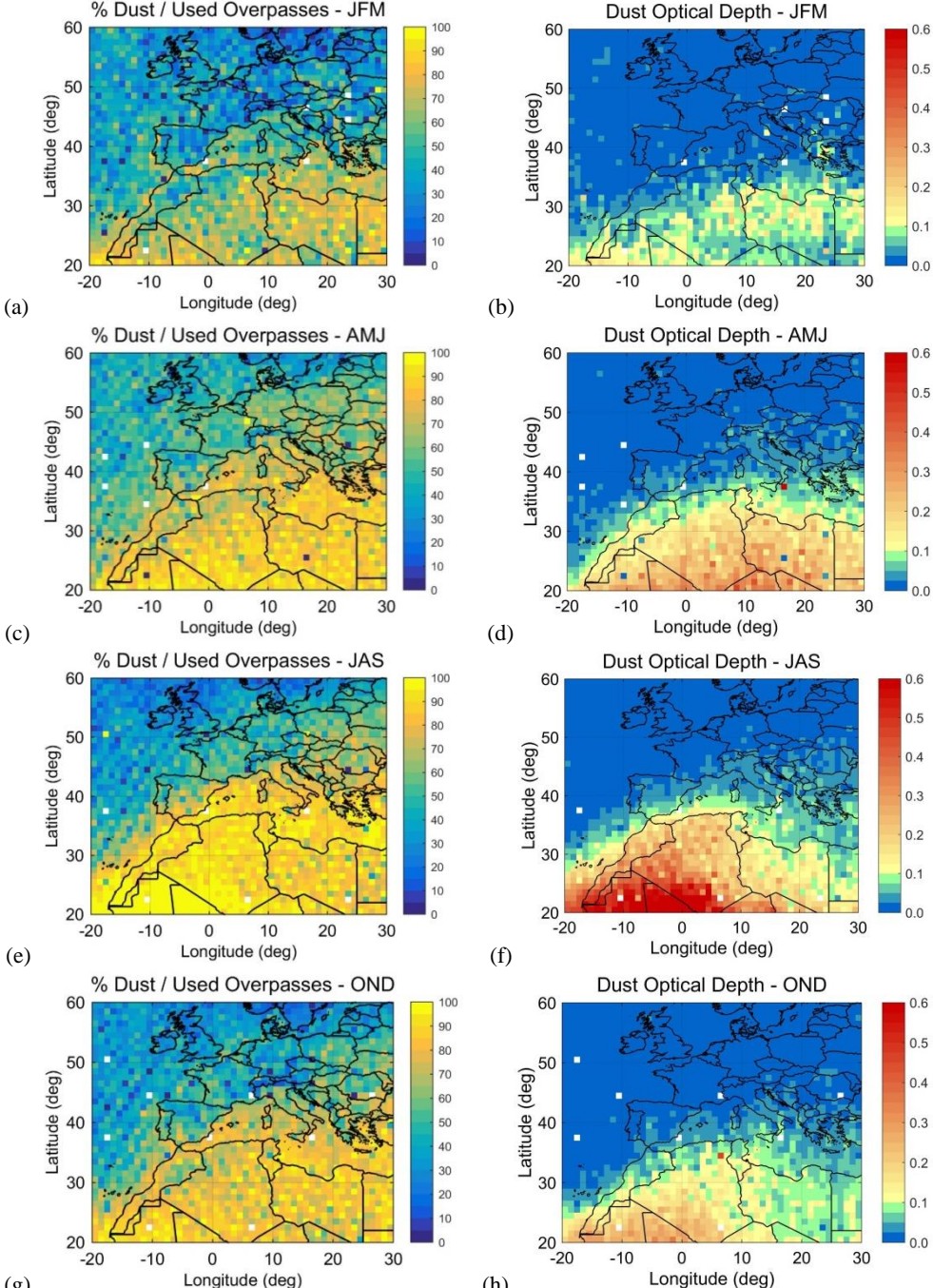

**Fig.1**

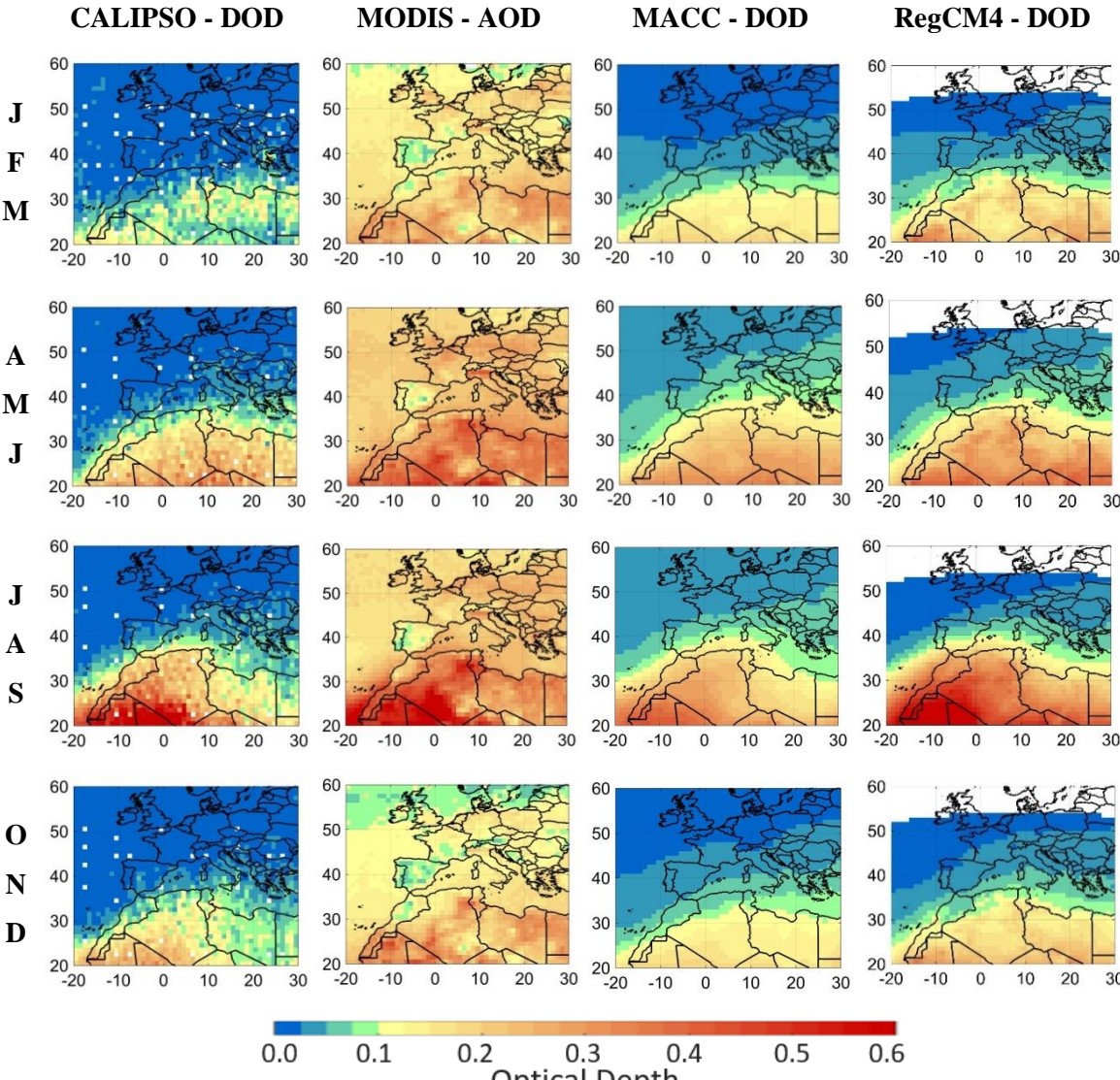

**Fig.2**

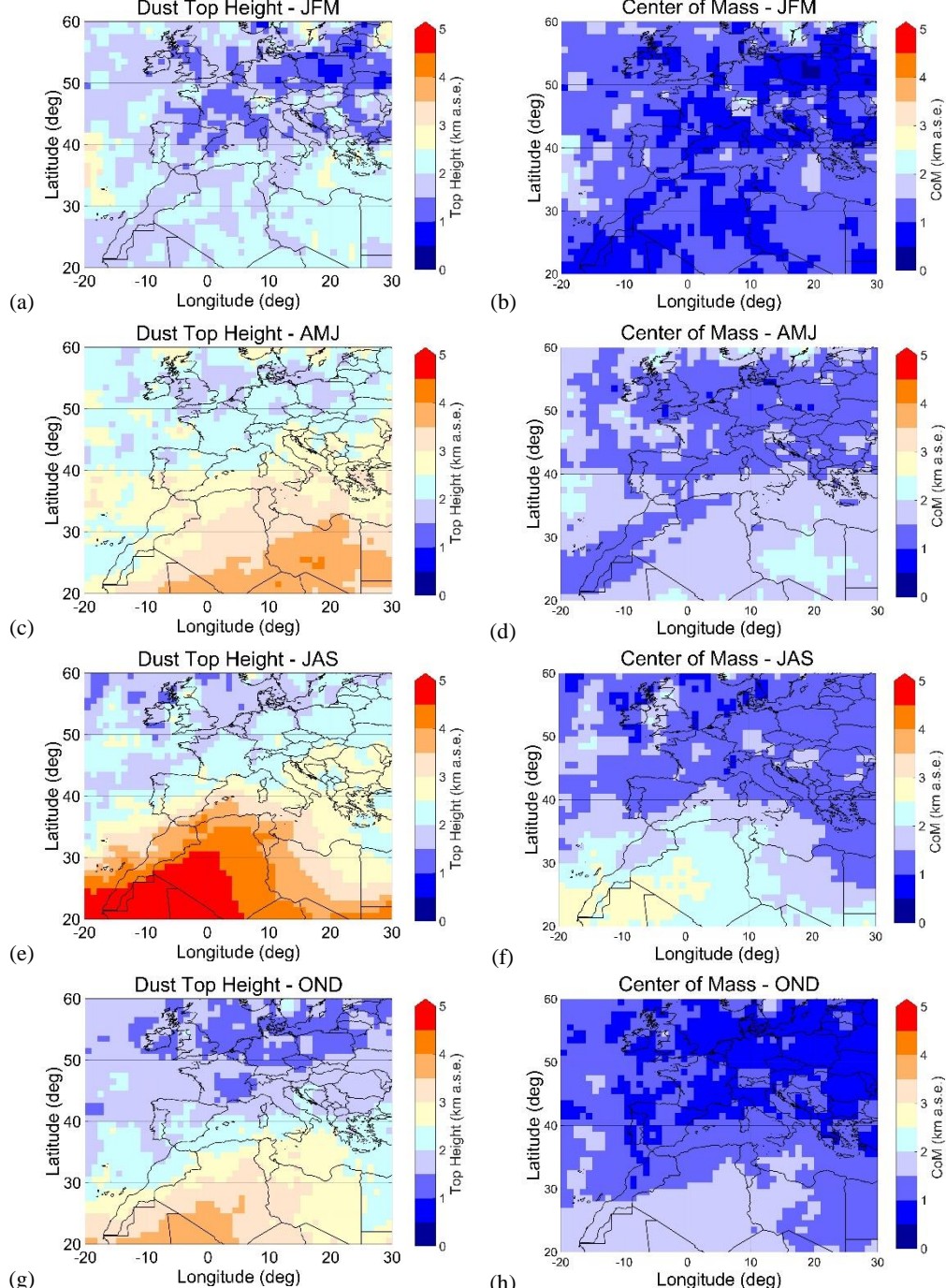

**Fig.3**

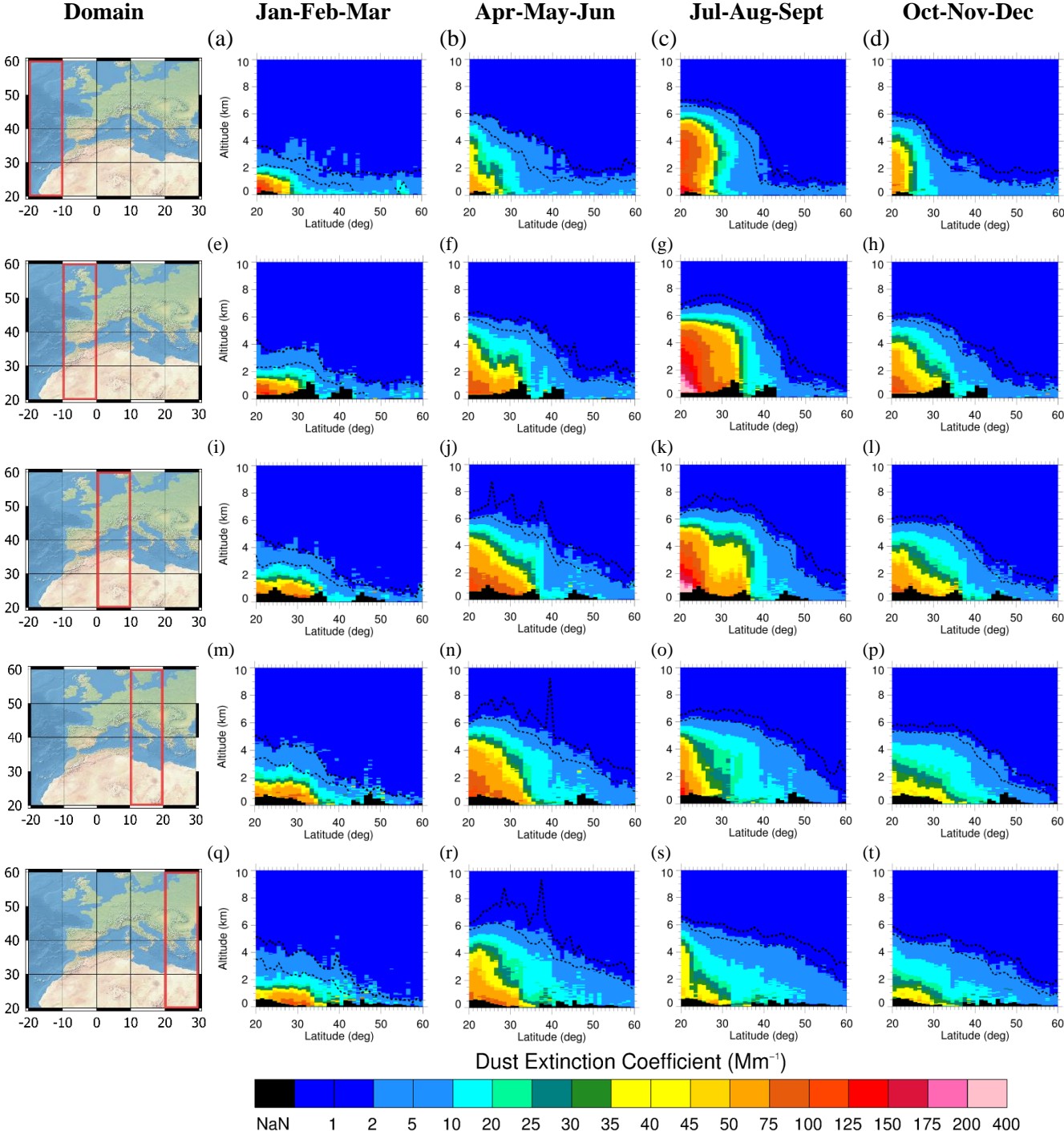

**Fig.4**

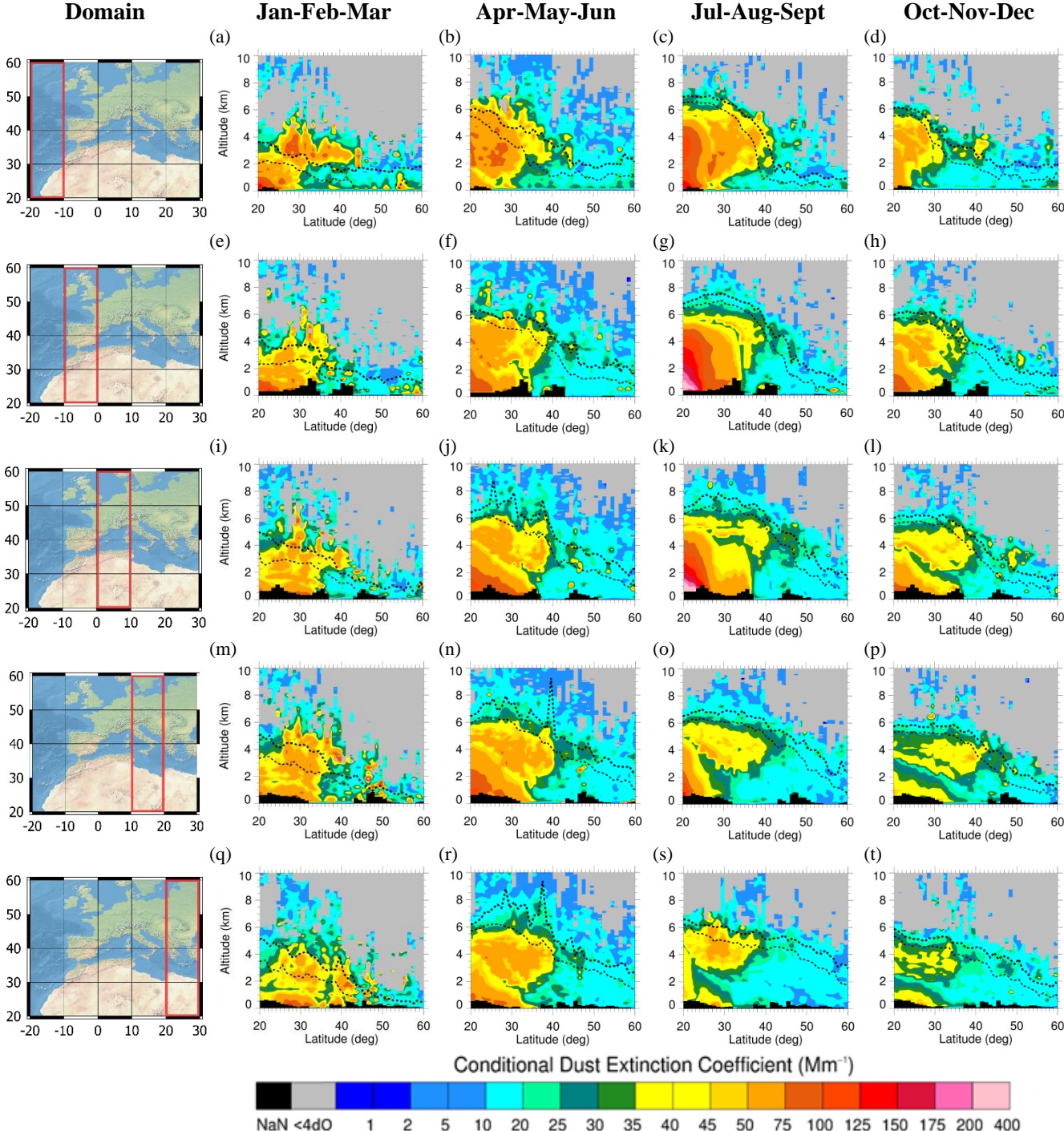

**Fig. 5**

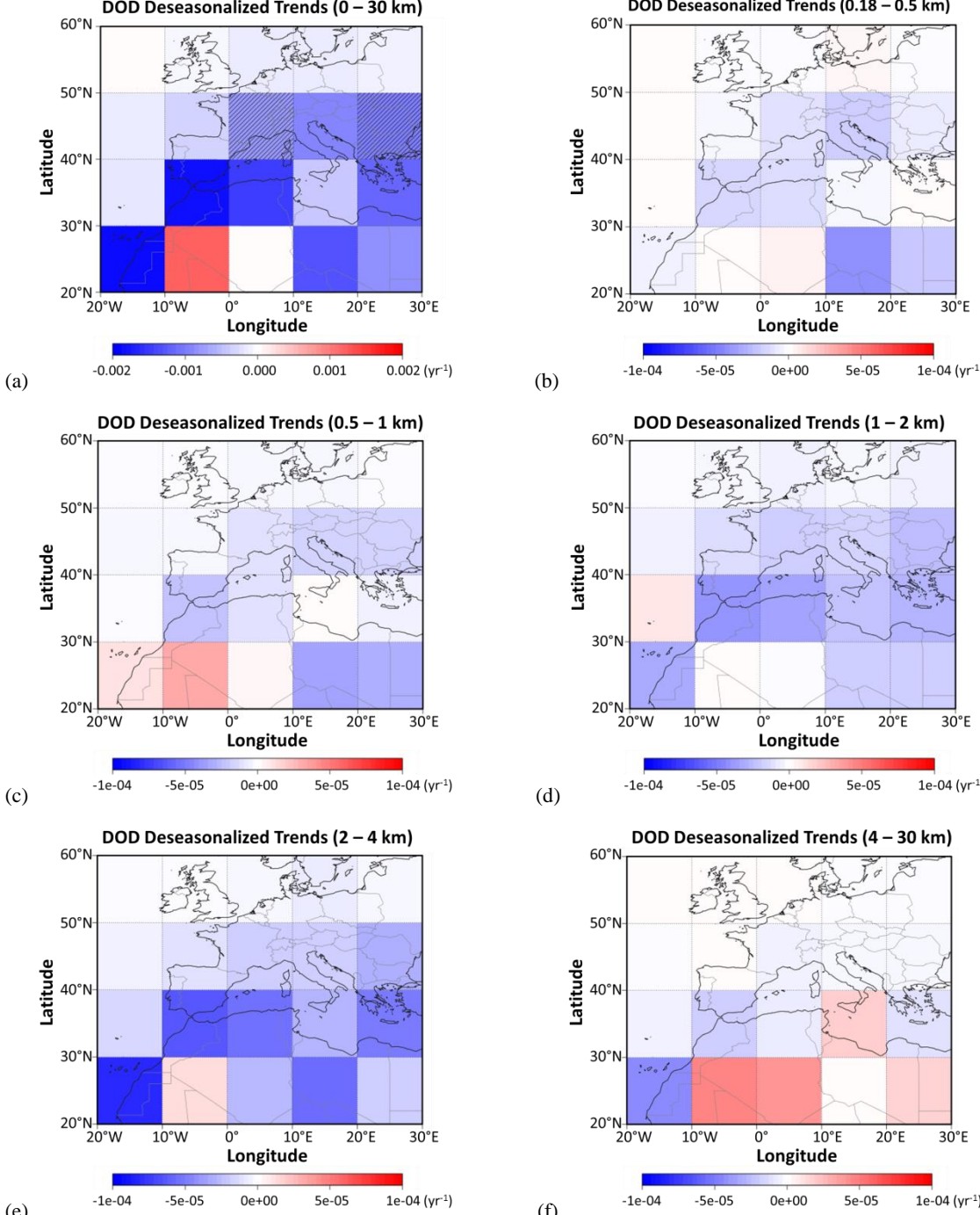

**Fig.6**

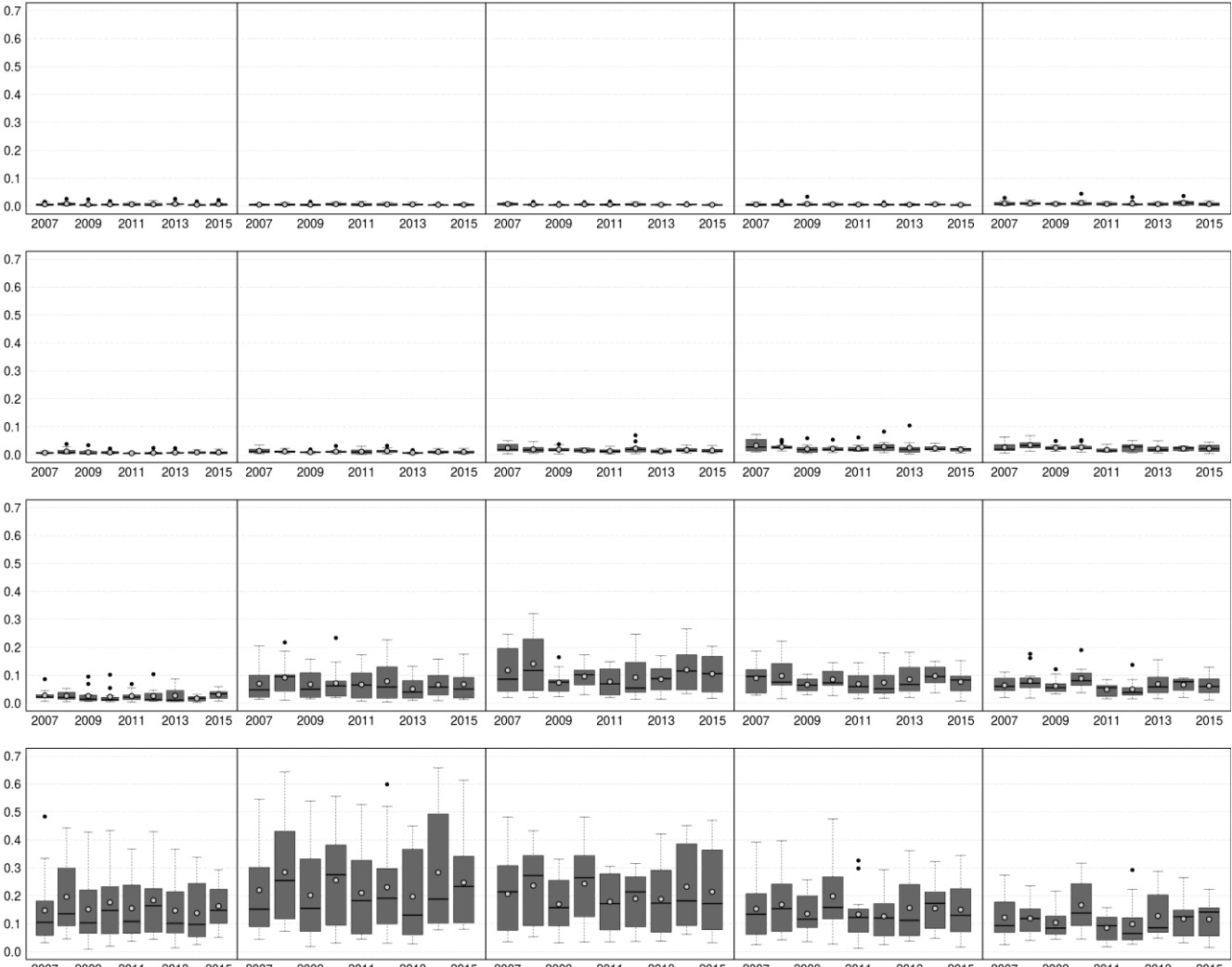

**Fig. 7**