# Peer review of "3D evolution of Saharan dust transport towards Europe based on a 9year EARLINET-optimized CALIPSO dataset"

_Atmospheric Chemistry and Physics, 2016_

## Referee Comment (RC1) · Anonymous Referee #3 · 21 Dec 2016

In this manuscript, "3D evolution of Saharan dust transport towards Europe based on a 9-year EARLINET-optimized CALIPSO dataset", the authors use a combination of CALIPSO and EARLINET to present a climatology of recent dust vertical distribution and transport to Europe from Africa. The consideration of both climatological extinction and 'conditional' extinction is useful to elucidate the episodic transport as well as the dust distribution.

The manuscript is generally well written and the climatology will be useful to the community and to evaluate models. However, there is a lack of discussion of uncertainties in the product and some limited interpretation of the particle depolarization ratio and interannual variability that need revision. Please see the major and minor comments

below.

Major Comments

There is limited discussion of the uncertainties and detection limits of dust occurrences throughout the manuscript. There are some very high occurrences of dust shown in regions far from dust sources in Figure 1 (e.g. the North Atlantic). How certain are we that this is actually dust and based on what detection limit? Similarly, Figures 3 and 4 show infrequent but high extinction dust at the surface at high latitudes. Can we be sure this is not a retrieval artifact? When climatological extinctions as low as 5 Mm-1 are considered (e.g. pg9 line 22) it would be useful to know the uncertainty on the estimates.

The retrieval is provided only for clear-sky conditions. Can you comment on how this might bias the dust extinction and how it relates to cloud formation (mentioned on page 10)?

In Figure 1 there is a strong boundary along the European coastline for dust occurrences, is this the result of a marked difference in used overpasses between the mainland and the Mediterranean? Please make sure that this feature is explained.

I don't think simply listing papers that have used specific instruments to explore dust over the Mediterranean is the best way of presenting the introduction (pg3 lines 10-20). Please consider reconstructing this paragraph to briefly discuss what these papers show that is relevant to understanding dust transport to Europe, rather than framing around the instrument used.

The seasonal climatological and conditional meridional dust extinction product will be useful for evaluating model representation of dust transport to Europe. I recommend that the authors make this available to the research community and include a link to the dataset in the manuscript, if possible.

The section on interannual variability is quite weak. The comparisons with other studies

should be relevant to the time period considered in this work. There does seem to be a general downward trend, but based on the lack of significance in many regions it is understandably difficult to determine long term trends over a relatively short 8 year period. Maybe the authors could include a timeseries panel in Figure 6 to indicate the interannual variability, rather than focus on the weak trends.?

Minor Comments

Please replace all instances of 'utilize' with 'use'

pg1 ln21 - "During spring..." sentence is not clear, please revise.

pg1 ln22 - "on" should be "in"

pg1 ln23 - "0.1", should this be "up to 0.1"?

pg1 ln28 - units are sometimes italicized, other times not

pg1 ln31 - change to "the Alps and Carpathian Mountains"

pg2 ln25 - remove "now"

pg3 ln30 - what is meant by "large scale statistics"

pg4 ln22 - extra space after "biases"

pg5 ln13 - perhaps provide the link as a reference?

pg5 ln21 - "categorized"

pg6 ln17 - "However..." it is not clear why this is an issue. Please elaborate or remove.

pg7 ln8 - "suppressed", this should be caveated as there are still significant emissions from African regions, like the Bodele, that are just not transported northwards.

pg7 ln23 - "Strong topographical heights", unclear meaning - please rephrase

pg8 ln3-5 - It is not clear what this tells us (high DOD, high sdev). Please explain what

this indicates.

pg8 ln3 - "In general,"

pg8 ln31 - "situation", please be more specific.

pg9 ln11 - "England" should be "Ireland"

pg9 ln15 - "England" should be "British Isles"

pg10 ln2 - "higher" than what?

pg10 ln18 - no new paragraph and replace "Nevertheless" with "However"

pg10 ln23 - delete the first sentence

pg10 ln24 - "Number of Exceedances", exceedances of what? Perhaps "Number of occurences" or "Number of observations" makes more sense?

pg11 ln3 - "move" should be "moves"

pg11 ln11 - "mean" should be "means"

pg11 ln11-20 - this section is out of place as the following paragraph returns to Fig. 4. Also, sentences in the paragraph somewhat contradict each other. If the PDR is a means of estimating age, but "cannot be considered as a possible age index". If the latter is true, why is this useful? The paragraph needs moving and restructuring, or removing (which would also mean removing the figure) unless the the PDR provides some insight.

pg11 ln22 - I think panel "l" should be panel "i"

pg11 ln25 - "plums" should be "plumes"

pg11 ln26 - give the latitude range of the mountainous regions

pg11 ln29 - "dust in" should be "dust at"

[Figure]

pg11 ln30 - can you be more specific why the deposition is stronger during that season - is it primarily wet or dry deposition?

pg11 ln31-34 - why is there a sudden drop off in extinction at 40N during the AMJ season? Please explain this.

pg12 ln1 - I think panel "i" should be panel "l"

pg12 ln19-24 - the studies referenced consider longer time periods and/or different geographical regions, please alter to so that the discussion relates better to the region and period you are considering.

pg12 ln26 - replace LR with lidar ratio

pg24 - The EARLINET reference is repeated multiple times.

Figure 2(b,d,f,h) - Why is the color bar different for the CoM panels relative to the Top Height panels when they are both showing altitude? Consider using the same color to avoid confusion

Figure 2 - In titles, "TOP" should be "Top"

Figures 3,4,5 - longitude and latitude labels are too small on the domain panels

---

## Referee Comment (RC2) · Anonymous Referee #4 · 22 Dec 2016

General remarks:

The present manuscript provides a monthly climatology (from 2007 to 2015) of African dust based on an optimised CALIPSO dust product was recently developed with a regional correction of the Saharan dust LR using EARLINET measurements (Amiridis et al., 2013). The monthly climatology of African dust obtained allows the description of the spatiotemporal features of dust properties over North Africa and Europe. The study of the mean state climatology shows strong seasonal shifts in dust source regions and transportation pathways. While the results of the study are interesting to be published, their presentation and discussion are not yet sufficient to be published in Atmospheric Chemistry and Physics in the current form. Therefore, it is worth to be published after

addressing major revisions which are explained below along with a few other details.

Major comments:

In Amiridis et al. (2013), this EARLINET-optimized CALIPSO dust optical depth (for the period 2007-2011) is described and qualitatively compared with MODIS and AERONET. The present manuscript is focusing on the analysis of the resulting EARLINET-optimized CALIPSO dust climatology. I would be desirable to include a short discussion of the uncertainties of the EARLINET-optimized CALIPSO dust product. I understand that this discussion is partly in Amiridis et al. (2013, 2015) although the authors should include a summary in Sect. 2.2 as well as about the uncertainties of the algorithm of CALIOP to determine the corresponding aerosol subtype (in Sect 2.1). In Figure 1, there are some features that they look associated with the number of available observations, and consequently with the presence of clouds over the Mediterranean and Europe. I am not sure if the "%Dust/Used Overpasses" is enough to explain the DOD seasonal patterns in Europe. I would suggest to include an additional column with the number of used overpasses and to check how is working the algorithm of CALIOP to determine the corresponding aerosol subtype in this part of the domain. In Page 10 Line 1, you mention that the results from Clim-DE can be used to estimate the impact of dust on cloud formation. As far as I understood, the EARLINET-optimized CALIPSO dust product is provided only for clear-sky conditions. In Sect. 4, you mention a recent paper from Mamouri and Ansmann (2016), but it is based on a ground-based lidar. Then, how could you estimate the dust impact on cloud formation from this EARLINET-optimized CALIPSO dust product? In my opinion, a further discussion about the similarities and discrepancies with other dust climatologies will enhance the impact of the results presented in the manuscript. Any comparison with other dust climatologies based on other datasets such as satellites (e.g. MODIS, AERONET, EARLINET or the official CALIPSO aerosol product); and models (as CAMS reanalysis or AEROCOM) is considered in the manuscript. Furthermore, how do the results of the present study improve those results of LIVAS (Amiridis et al., 2015)? These discussions will be useful for model evaluation, for example. Otherwise, it seems to me that some results are general and not enough justify in the manuscript. In Sect. 3.1 (Page 10) I don't understand the reason to include the dust mass concentration inversion results. This part of the discussion doesn't include any new insight with respect the analysis of the optical properties or any link to a particular previous study. In Sect. 3.5, you could compare your results with a climatic index as the North Atlantic Oscillation Index (NAO) as Pey et at. (2013) did for PM10. Is this de-seasonalised trend analysis sensitive to the number of available observations?

Minor comments:

Page 2 Line 13: Add Nickovic et al. (2016).

Page 2 Line 16: Add Granados-Muñoz et al. (2016) and Bovchaliuk et al. (2016).

Page 3 Line 25: Replace Gkikas et al. (2015, ACPD) by Gkikas et al. (2016, ACP).

Page 3 Line 30: When you said "large scale statistics of discriminate and optimised dust extinction and AOD fields from CALIPSO", what does it mean? What about Amiridis et al. (2013) and Amiridis et al. (2015)?

Sect. 3.1: It would be good if you can add a short comparison of the resulting DOD seasonal maps with the results of MODIS, MISR or any available reanalysis (as CAMS or MERRA).

Sect. 3.2: In this case, you could compare your results from those obtained from EARLINET o models.

Sect 3.5: how do your results fit with those showed in Gkikas et al. (2016)?

Page 10 Line 28: Add Huneeus et al. (2016).

Page 10 Line 30: "it is likely that the surface and elevated dust have different origins" sounds speculative. You could check this assumption with models or back trajectories.

Figure 3. I would use the same colour palette than in Figure 4. Moreover, could you provide any further explanation about the sharp transition over the Atlas?

Figures 3,4,5: Longitude and Latitude labels can be removed. They are too small.

References:

Amiridis, V., Marinou, E., Tsekeri, A., Wandinger, U., Schwarz, A., Giannakaki, E., Mamouri, R., Kokkalis, P., Binietoglou, I., Solomos, S., Herekakis, T., Kazadzis, S., Gerasopoulos, E., Proestakis, E., Kottas, M., Balis, D., Papayannis, A., Kontoes, C., Kourtidis, K., Papagiannopoulos, N., Mona, L., Pappalardo, G., Le Rille, O., and Ansmann, A.: LIVAS: a 3-D multi-wavelength aerosol/cloud database based on CALIPSO and EARLINET, Atmos. Chem. Phys., 15, 7127-7153, doi:10.5194/acp-15-7127-2015, 2015.

Bovchaliuk, V., Goloub, P., Podvin, T., Veselovskii, I., Tanre, D., Chaikovsky, A., Dubovik, O., Mortier, A., Lopatin, A., Korenskiy, M., and Victori, S.: Comparison of aerosol properties retrieved using GARRLiC, LIRIC, and Raman algorithms applied to multi-wavelength lidar and sun/sky-photometer data, Atmos. Meas. Tech., 9, 3391-3405, doi:10.5194/amt-9-3391-2016, 2016.

Gkikas, A., Basart, S., Hatzianastassiou, N., Marinou, E., Amiridis, V., Kazadzis, S., Pey, J., Querol, X., Jorba, O., Gassó, S., and Baldasano, J. M.: Mediterranean intense desert dust outbreaks and their vertical structure based on remote sensing data, Atmos. Chem. Phys., 16, 8609-8642, doi:10.5194/acp-16-8609-2016, 2016.

Granados-Muñoz, M. J., Navas-Guzmán, F., Guerrero-Rascado, J. L., Bravo-Aranda, J. A., Binietoglou, I., Pereira, S. N., Basart, S., Baldasano, J. M., Belegante, L., Chaikovsky, A., Comerón, A., D'Amico, G., Dubovik, O., Ilic, L., Kokkalis, P., Muñoz-Porcar, C., Nickovic, S., Nicolae, D., Olmo, F. J., Papayannis, A., Pappalardo, G., Rodríguez, A., Schepanski, K., Sicard, M., Vukovic, A., Wandinger, U., Dulac, F., and Alados-Arboledas, L.: Profiling of aerosol microphysical properties at several

EARLINET/AERONET sites during the July 2012 ChArMEx/EMEP campaign, Atmos. Chem. Phys., 16, 7043-7066, doi:10.5194/acp-16-7043-2016, 2016.

Huneeus, N., Basart, S., Fiedler, S., Morcrette, J.-J., Benedetti, A., Mulcahy, J., Terradellas, E., Pérez García-Pando, C., Pejanovic, G., Nickovic, S., Arsenovic, P., Schulz, M., Cuevas, E., Baldasano, J. M., Pey, J., Remy, S., and Cvetkovic, B.: Forecasting the northern African dust outbreak towards Europe in April 2011: a model intercomparison, Atmos. Chem. Phys., 16, 4967-4986, doi:10.5194/acp-16-4967-2016, 2016.

Nickovic, S., Cvetkovic, B., Madonna, F., Rosoldi, M., Pejanovic, G., Petkovic, S., and Nikolic, J. (2016) Cloud ice caused by atmospheric mineral dust – Part 1: Parameterization of ice nuclei concentration in the NMME-DREAM model" Atmos. Chem. Phys., 16, 11367-11378, doi:10.5194/acp-16-11367-2016.
* * *

---

## Referee Comment (RC3) · Anonymous Referee #2 · 11 Jan 2017

The paper entitled "3D evolution of Saharan dust transport towards Europe based on a 9-year EARLINET-optimized CALIPSO dataset" is an interesting analysis of mineral dust properties above North Africa, the Mediterranean and Europe that contains valuable information in 3 dimensions using CALIPSO products improved with EARLINET techniques and data.

However, the manuscript needs to undergo some improvements before being published in ACP. First, I suggest to improve the English and writing throughout the manuscript. Additionally, results presented here are valuable and interesting but in general discussion need to be extended and completed at some points in Section 3. I suggest that the authors include more statistics such as the mean, standard deviation, extreme values, etc for some of the properties presented here and for the different regions. Some sentences comparing the results obtained in Section 3 with results obtained in previous studies would also be useful. They should also consider the use of tables to summarize main results, making easier for the reader to focus on the main findings of the study. Consider also the minor comments following next:

I suggest to replace the word utilize by use

Page 2, line 26: Replace "means of identifying" by "mean of identifying"

Page 2, line 29: Remove "a" before pure dust extinction

Page 2, line 31: Replace later by latter

Page 3, line 17-18: Is the climatology by Winker et al, 2013 on dust properties? If not, remove it from the paragraph

Page 4, line 24: Replace CALISPO by CALIPSO

Page 4, line 27: Explain the acronym LIVAS

Page 5, line 5: Did you quantify this error? Could you provide an estimated value here?

Page 5, line 6: I suggest replacing "Based on this this technique" by "On using this technique"

Page 5, line 31: I suggest starting a new paragraph from "The conditional dust product..."

Page 6, line 9: What do you mean they should be used with caution? Because of the definition provided here, it is expected that Con-DE is larger than total extinction for some cases, but it is still correct

Page 6, lines 11-16: It will be useful to include in this paragraph the information about the region studied and the period covered

Page 6, line 28: Remove "of the" before "mean DOD values"

Page 6, line 30: Please add a short sentence here explaining why dust transport is suppressed

Page 7, line 17: Provide the precise value of the mean DOD and its standard deviation instead of ranges or rephrase the sentence

Page 8, line 4: Does $\alpha$ represent the total aerosol extinction or the dust aerosol extinction?

Page 8, line 22: Replace "situation" by "horizontal pattern" or "horizontal distribution"

Page 8, line 24: I suggest renaming section 3.3. as "Climatological dust cross sections" to be coherent with the title in section 3.4.

Page 9, line 3: what do you mean by mobilization of the sources here? Please, elaborate more this sentence

Figure 3 (4, and 5): Please, increase the size of the axis labels text for the Domain figures

Page 9, line 12-13: Elaborate this sentence

Page 9, line 15: Similar Clim-DE values are observed between 50-60 deg N for other longitudinal zones, why do you point it out for this specific zone? Also, what is the uncertainty for the Clim-DE product? Values of 5 Mm-1 are very low and could fall within the uncertainty. Add discussion regarding the uncertainty throughout the manuscript where needed

Page 9, line 16: What are the criteria to consider a value of 10 Mm-1 "significantly" high?

Page 9, line 29-33: You should consider adding here more discussion and some statistical parameters (e.g. mean, standard deviation, maxima, minima, etc) to enrich this summary. Also, some sentences about the dust vertical distribution in the summary are missing.

Page 10, line 1: How is the impact on cloud formation estimated?

Page 10, line 2: Please, include additional information and discussion on this part related to the dust mass concentration calculation. What is the point of calculating it here?

Page 10, lines 12-14: The information included here should be provided earlier in the section, before discussing the results.

Page 10, line 23: Replace "populations of dust" by "dust features"

Page 10, line 25: Indicate the other seasons and regions where the two distinct layers are observed

Page 11, lines 3-12: This paragraph should be moved to later on in the manuscript, in order to keep all the discussion related to figure 4 together. Additionally, more discussion on depolarization should be provided here.

Page 11, line 16: Replace "in the same range with" by "in the same range as"

Page 11, line 32: At the end of section 3.3 you mentioned that Con-De will be used to discuss if the decreasing intensity with height and latitude is representative, but this is not discussed in section 3.4. Please, include some sentences. Additionally, some more discussion comparing the results from sections 3.4 and 3.3 will be interesting.

Page 12, line 18: Replace "statistical significant" by "statistically significant"

---

## Author Comment (AC1) · 3 Mar 2017

**The paper entitled "3D evolution of Saharan dust transport towards Europe based on a 9-year EARLINET-optimized CALIPSO dataset" is an interesting analysis of mineral dust properties above North Africa, the Mediterranean and Europe that contains valuable information in 3 dimensions using CALIPSO products improved with EARLINET techniques and data. However, the manuscript needs to undergo some improvements before being published in ACP.**

[REPLY] We thank the reviewer for the thorough revision and comments. Replies to the general and specific comments follow below.

**General comments**

**First, I suggest to improve the English and writing throughout the manuscript.**

[REPLY] We have revised the manuscript for language issues.

**Additionally, results presented here are valuable and interesting but in general discussion need to be extended and completed at some points in Section 3. I suggest that the authors include more statistics such as the mean, standard deviation, extreme values, etc for some of the properties presented here and for the different regions.**

[REPLY] We thank the reviewer for his suggestion. We revised Section 3 by including the discussion of DOD statistics (mean, standard deviation and extreme values - Section 3.1), the dust heights (standard deviations of CoM and TH – Section 3.2) and statistic on the extinction coefficient values (mean and Standard deviation - Section 3.3). The new discussions are the following:

Page 9, lines: 27: "More specific, during JFM (Figs. 1a, b) limited dust activity is observed almost uniformly over the Sahara desert. The DOD remains roughly over the entire study domain below 0.13 with 75% of the observations having DODs < 0.17, 95% of the observation having DODs < 0.5 and extreme values with DODs ~2."

Page 10, lines 1: "In the domains between 10° E - 30° E and 30° N - 40° N, 5% of the dust events are observed with DODs > 0.41, 1% with DODs >0.95 and extreme observations with DODs are up to 1.6."

Page 10, lines 7: "During AMJ (Figs 1c, d) dust production occurring over the entire Saharan desert with mean DOD values of 0.26 ± 0.26 and occurrences of 86%, uniformly at latitudes between 20° N and 30° N."

Page 10, lines 16:" In the domain between 10° W - 00° and 20° N - 35° N, the mean DOD is 0.43, with 25% of the dust observations having DODs > 0.69, 5% >1.2 and the extreme DODs up to 3 (Table 2)."

Page 10, lines 20 : "In the domain between 10° W - 00° and 35° N - 45° N, the mean DODs are 0.09 ±,0.14 with 5% of the dust observations having DODs >0.55 and extremes DODs up to 2.3."

Page 11, lines 16: "During JFM dust resides in general below 3 $km$ a.s.e. (above surface elevation) over land with CoM at about 1.3 ± 1.6 $km$ a.s.e. (Figs. 3a, b). Over the sea, several transport paths are discernible especially over eastern Mediterranean with dust tops traveling at 2.3 ± 1.9 $km$ a.s.e. During AMJ, TH and CoM are up to 4.2 ± 1.7 $km$ and around 2.4 ± 1.1 $km$ a.s.e. respectively over eastern parts of Sahara."

Page 11, lines 26: "This pattern leads to elevated dust at 3.0 ± 1.7 $km$ a.s.e. and CoM at 1.6 ± 1.1 $km$ a.s.e. over south European countries and Balkans. During OND the horizontal pattern is similar to JJA however with much lower heights (Figs. 3g, h)."

Page 13, lines 10: "Above the Balkans and during JFM values of 29 ± 65 $Mm^{-1}$ are observed in the first 1.5 $km$, and 10 ± 30 $Mm^{-1}$ between 2.5 – 3.5 $km$. In AMJ and JAS respectively, means of ~ 16 ± 40 $Mm^{-1}$ and ~ 9 ± 20 $Mm^{-1}$ are observed in altitudes between 1.5 to 5 $km$. The values of Clim-DE are higher (>45 $Mm^{-1}$) over Africa during winter and spring, in relation with the ones observed during the other two seasons (<45 $Mm^{-1}$) and reach high altitudes (5-6 $km$ a.s.l.) during spring and summer. In summary, the obtained cross-sections for the five longitudinal zones indicate that higher extinction coefficient values are observed near the source and at low altitudes, where dust particles are efficiently deposited. Above NE Africa, the Clim-DE values are >45 $Mm^{-1}$ throughout the year in altitudes up to 2 $km$ a.s.l. during JFM and up to 4 km during AMJ and JJA. Moreover, the standard deviation of the means is around 130% at the altitudes up to 2 km and ~100% between 2 – 4 km, at all seasons. Above West Africa, the extreme Clim-DE values observed during JAS in the altitudes up to 2 km are 113 ± 131 $Mm^{-1}$. In C-E Mediterranean, dust is always present, with maximum extinctions during AMJ, reaching 27 ± 54 $Mm^{-1}$ close to the surface and ~ 18 ± 30 $Mm^{-1}$ during JAS and OND. In C-W Mediterranean, the highest means of JAS are ~16 ± 40 $Mm^{-1}$. For latitudes greater than 45° N, and during AMJ mean values of  8 ± 27 $Mm^{-1}$ are  4 ± 16 $Mm^{-1}$ are observed close to the surface above NE Europe and NW Europe respectively."

**Some sentences comparing the results obtained in Section 3 with results obtained in previous studies would also be useful.**

[REPLY] We revised Section 3 by including discussion on the comparison of the results obtained in this work with results in previews studies (Papayannis et al. 2008; Balis et al. 2012; Mona et al. 2014). In section 3.2 we included a comparison with the dust plume heights documented by EARLINET. In Section 3.5 we included comparison of our trend with other studies over the same domain (Floutsi et al. 2016; Gkikas et al. 2013; Yoon et al. 2012; Georgoulias et al. 2016b). The new sections are:

Page 11, line 29: "In general, our results are in agreement with lidar-based studies which have been performed in several European sites. Papayannis et al. (2008) performed an exhaustive analysis on Saharan dust particles over Europe using EARLINET lidar profiles. They found that the dust layer center of mass extends from 3.0 to 3.8 km and the thickness ranges from 0.7 to 3.4 km. Specifically, Balis et al. (2012) calculated the mean base and top of dust layers in the eastern Mediterranean, Thessaloniki, to be around 2.5 ± 0.9 $km$ and 4.2 ± 1.5 $km$, respectively. More recently, Mona et al. (2014) analyzed a long dataset of Saharan dust intrusions over Potenza, Italy, and found a mean layer centre of mass of 3.5 ± 1.5 $km$."

Page 15, line 29: "In comparison with studies relevant to the time period considered in this work, the DOD decrease of 0.001 $yr^{-1}$ over the northern coast of Africa is in agreement with Floutsi et al. (2016), who based on 12 years of MODIS-Aqua observations (2002-2014) reported an average decrease of 0.003 $yr^{-1}$ for the coarse mode fraction of AOD over the broader Mediterranean Sea. Furthermore, over the same domain the decreasing trend of DOD coincides with the decrease of Saharan desert dust episodes as reported by Gkikas et al. (2013). Regarding the AERONET stations over the domain of northern Africa and Europe, Yoon et al. (2012) reported on the trends of AOD at 440 nm along with the corresponding Ångström Exponents (440 and 870nm). The documented negative trends over the AERONET stations of Avignon (France), Dakar (Senegal) and Ispra (Italy) are in agreement with the negative DOD reported here, although with discrepancies in the magnitude, while trend disagreements are observed over the AERONET station of Banizoumbou (Niger). The decreasing trends of DOD observed over the domain northern of Africa and Europe coincide with the generally documented downward AOD trends reported based on several satellite observations of MODIS/Aqua, MODIS/Terra, MISR and SeaWiFS (Pozzer et al., 2015; de Meij et al., 2012; Hsu et al., 2012; Georgoulias et al. 2016b). More particular, in the most recent study of Georgoulias et al. (2016b), using MODIS/Terra and MODIS/Agua observations, they reported negative statistically significant trends over Algeria, Egypt and the Mediterranean and positive trends over Middle East. Overall, for the Mediterranean they reported an AOD trend of -0.0008 $yr^{-1}$ for the MODIS/Terra observations (2000 – 2015) and -0.0020 $yr^{-1}$ for the MODIS/Aqua observations (2002 – 2015), with the trends being statistical significant at the 95% confidence level in both cases."

**They should also consider the use of tables to summarize main results, making easier for the reader to focus on the main findings of the study.**

[REPLY] We introduced a new Table 2 where we summarize main results for different regions and seasons. We agree with the reviewer that this will help the reader to focus on our main findings. The new Table 2 is (page 37):

**Table 2: Regional statistics on mean dust optical depth, max values, dust layer center of mass (CoM) and top height (TH) (a. s. e.), ratio of dust observations to cloud-free observations, ratio of cloud-free observations to total observations and domain boundaries.**

| | DOD Mean ± St.dev. | DOD Max Vals. (Perc. 95%) | CoM ± St.dev. | Top Height ± St.dev. | Nr Dst in Nr cl-free | Nr cl-free in Nr obs. | Domain |
|---|---|---|---|---|---|---|---|
| **NE Africa** | | | | | | | |
| JFM | 0.11 ± 0.17 | 2.19 (0.42) | 1.5 ± 1.2 | 2.6 ± 1.8 | 0.72 | 0.84 | [10E,30E] |
| AMJ | 0.26 ± 0.26 | 3.09 (0.73) | 2.4 ± 1.1 | 4.2 ± 1.7 | 0.86 | 0.86 | [20N,30N] |
| JAS | 0.18 ± 0.21 | 2.63 (0.56) | 2.3 ± 1.0 | 4.0 ± 1.4 | 0.84 | 0.93 | |
| OND | 0.11 ± 0.14 | 2.93 (0.34) | 1.9 ± 0.9 | 3.3 ± 1.4 | 0.81 | 0.93 | |
| **NW Africa** | | | | | | | |
| JFM | 0.13 ± 0.18 | 1.86 (0.47) | 1.5 ± 1.3 | 2.4 ± 1.8 | 0.67 | 0.82 | [10W,10E] |
| AMJ | 0.26 ± 0.26 | 2.31 (0.75) | 2.2 ± 1.2 | 3.8 ± 1.6 | 0.86 | 0.83 | [20N,35N] |
| JAS | 0.43 ± 0.39 | 3.03 (1.20) | 2.9 ± 1.0 | 5.1 ± 1.3 | 0.94 | 0.88 | |
| OND | 0.22 ± 0.26 | 2.59 (0.71) | 2.2 ± 1.0 | 3.9 ± 1.6 | 0.82 | 0.81 | |
| **C-E Med.** | | | | | | | |
| JFM | 0.09 ± 0.18 | 1.62 (0.41) | 1.3 ± 1.4 | 2.3 ± 1.9 | 0.69 | 0.70 | [10E,30E] |
| AMJ | 0.12 ± 0.20 | 2.74 (0.51) | 1.8 ± 1.5 | 3.2 ± 2.1 | 0.82 | 0.76 | [30N,45N] |
| JAS | 0.08 ± 0.12 | 1.80 (0.33) | 1.6 ± 1.1 | 3.0 ± 1.7 | 0.89 | 0.96 | |
| | 0.08 ± 0.11 | 1.55 (0.31) | 1.4 ± 1.1 | 2.7 ± 1.6 | 0.82 | 0.80 | |

| | | | | | | | |
|---|---|---|---|---|---|---|---|
| OND | | | | | | | |
| **C-W Med.** | | | | | | | [10W,10E] [35N,45N] |
| JFM | 0.03 ± 0.06 | 1.09 (0.11) | 1.3 ± 1.6 | 2.0 ± 1.9 | 0.49 | 0.57 | |
| AMJ | 0.05 ± 0.10 | 1.35 (0.25) | 1.8 ± 1.6 | 2.9 ± 2.2 | 0.65 | 0.61 | |
| JAS | 0.09 ± 0.14 | 2.33 (0.36) | 1.9 ± 1.2 | 3.3 ± 1.8 | 0.75 | 0.80 | |
| OND | 0.05 ± 0.09 | 1.62 (0.20) | 1.3 ± 1.2 | 2.3 ± 1.6 | 0.63 | 0.64 | |
| **NE Europe** | | | | | | | [10E,30E] [45N,60N] |
| JFM | 0.025 ± 0.055 | 0.97 (0.11) | 1.2 ± 1.4 | 1.7 ± 1.7 | 0.37 | 0.28 | |
| AMJ | 0.033 ± 0.062 | 1.61 (0.12) | 1.6 ± 1.2 | 2.5 ± 1.6 | 0.61 | 0.47 | |
| JAS | 0.032 ± 0.045 | 0.90 (0.11) | 1.6 ± 1.1 | 2.7 ± 1.4 | 0.60 | 0.58 | |
| OND | 0.023 ± 0.043 | 0.50 (0.09) | 1.2 ± 1.0 | 1.9 ± 1.4 | 0.49 | 0.43 | |
| **NW Europe** | | | | | | | [10W,10E] [45N,60N] |
| JFM | 0.015 ± 0.033 | 0.47 (0.06) | 1.2 ± 1.6 | 1.7 ± 1.7 | 0.36 | 0.36 | |
| AMJ | 0.023 ± 0.037 | 0.73 (0.08) | 1.5 ± 1.6 | 2.2 ± 1.9 | 0.52 | 0.47 | |
| JAS | 0.022 ± 0.042 | 0.93 (0.08) | 1.4 ± 1.5 | 2.1 ± 1.7 | 0.43 | 0.52 | |
| OND | 0.018 ± 0.035 | 0.57 (0.07) | 1.1 ± 1.2 | 1.7 ± 1.4 | 0.40 | 0.44 | |

**Detailed comments**

**I suggest to replace the word utilize by use**

[REPLY] It is replaced throughout the manuscript.

**Page 2, line 26: Replace "means of identifying" by "mean of identifying"**

[REPLY] It is replaced.

**Page 2, line 29: Remove "a" before pure dust extinction**

[REPLY] It is removed.

**Page 2, line 31: Replace later by latter**

[REPLY] It is replaced.

**Page 3, line 17-18: Is the climatology by Winker et al, 2013 on dust properties? If not, remove it from the paragraph**

[REPLY] We changed the sentence in order to clarify the contribution of this study:

Page 3, line 24: "Moreover, Winker et al. (2013) provided a 3D global aerosol climatology from five-year CALIPSO data, along with the global distribution of mineral dust, derived using the ratio of columnar dust AOD to total AOD."

**Page 4, line 24: Replace CALISPO by CALIPSO**

[REPLY] It is corrected.

**Page 4, line 27: Explain the acronym LIVAS**

[REPLY] We added the acronym's explanation:

Page 5, line 3: "This product is a prominent outcome from the EARLINET-ESA collaboration for the LIVAS database (LIdar climatology of Vertical Aerosol Structure for space-based lidar simulation studies; Amiridis et al., 2015)".

**Page 5, line 5: Did you quantify this error? Could you provide an estimated value here?**

[REPLY] We added the information in this sentence:

Page 5, lines 13: "During SAMUM 1 and 2 campaigns Saharan dust $\delta_{nd}$ values varied between 0.27 and 0.35 at 532 nm (Ansmann et al., 2011), introducing 4% error in our calculations for the dust separated backscatter values."

**Page 5, line 6: I suggest replacing "Based on this this technique" by "On using this technique"**

[REPLY] The sentence is rephrased.

**Page 5, line 31: I suggest starting a new paragraph from "The conditional dust product: : :"**

[REPLY] Done.

**Page 6, line 9: What do you mean they should be used with caution? Because of the definition provided here, it is expected that Con-DE is larger than total extinction for some cases, but it is still correct**

[REPLY] This sentence has been removed from the revised manuscript.

**Page 6, lines 11-16: It will be useful to include in this paragraph the information about the region studied and the period covered**

[REPLY] We changed the first sentence as:

Page 8, line 21: "In Sect. 3.1 - 3.4, we examine the inter-seasonal variation and intensity of dust transport patterns, from 2007 to 2015, for the domain 20° W to 30° E and 20° N to 60° N."

**Page 6, line 28: Remove "of the" before "mean DOD values"**

[REPLY] It is removed.

**Page 6, line 30: Please add a short sentence here explaining why dust transport is suppressed**

[REPLY] We added the sentence:

Page 9, line 26: "During autumn and winter the emission and transport of dust towards Europe is suppressed due to the more effective removal processes and due to the atmospheric dynamics favouring the transport of dust towards the Atlantic (e.g. Israelevich et al., 2002; Schepanski et al., 2009)."

**Page 7, line 17: Provide the precise value of the mean DOD and its standard deviation instead of ranges or rephrase the sentence**

[REPLY] We rephrased the sentence:

"Mean DOD over these areas reaches values of 0.12 ± 0.20 (Fig. 1d) and extreme observations observed with DODs up to 2.74."

**Page 8, line 4: Does represent the total aerosol extinction or the dust aerosol extinction?**

[REPLY] It represents the dust extinction. We improved the sentence: "α denotes the dust extinction coefficient at altitude z."

**Page 8, line 22: Replace "situation" by "horizontal pattern" or "horizontal distribution"**

[REPLY] We rephrased as:

Page 11, line 27: "During OND the horizontal pattern is similar to JJA however with much lower heights (Figs. 3g, h)."

**Page 8, line 24: I suggest renaming section 3.3. as "Climatological dust cross sections" to be coherent with the title in section 3.4.**

[REPLY] It has been replaced.

**Page 9, line 3: what do you mean by mobilization of the sources here? Please, elaborate more this sentence**

[REPLY] We changed the sentence as:

Page 13, line 13: "The spring and summer peaks indicate the increased activity of Saharan dust sources (Moulin et al., 1998; Schepanski et al., 2007)."

**Figure 3 (4, and 5): Please, increase the size of the axis labels text for the Domain figures**

[REPLY] The label size is increased and, now, it is more visible in the new version of the manuscript.

**Page 9, line 12-13: Elaborate this sentence**

[REPLY] We changed the sentence accordingly:

Page 12, line 23: "A steep decrease in extinction values is observed along the African coastline with values of 20 $Mm^{-1}$ above the southern part of the Iberian Peninsula (38°-42° N) where dust is trapped by the Pyrenees. The distinct decrease of extinction values across the African coastline is an indication that dust is always present inside the rather deep Saharan boundary layer while it is only occasionally transferred towards the Mediterranean when atmospheric dynamics favor this kind of flow."

**Page 9, line 15: Similar Clim-DE values are observed between 50-60 deg N for other longitudinal zones, why do you point it out for this specific zone? Also, what is the uncertainty for the Clim-DE product? Values of 5 Mm-1 are very low and could fall within the uncertainty. Add discussion regarding the uncertainty throughout the manuscript where needed**

[REPLY] We changed the sentence:

Page 12, line 27: "At higher latitudes, the CALIPSO dust extinction is drastically reduced but still observed in ranges of 1-2 $km$ a.s.l. and with mean Clim-DE values of 5 $Mm^{-1}$."

In Clim-DE and Cond-DE products, the uncertainty of the dust extinction values close to the surface and at high latitudes are < 54%. At high altitudes and for latitudes up to 45°N, the uncertainty of the values is < 20%. We added this in the manuscript in the new section 2.4 addreses the uncertainties of the product:

Page 7, line 26: "In general, Clim-DE and Cond-DE products, the uncertainty of the dust extinction values close to the surface and at high latitudes is < 54%. At high altitudes and for latitudes up to 45°N, the uncertainty of the values is < 20%."

**Page 9, line 16: What are the criteria to consider a value of 10 Mm-1 "significantly"high?**

[REPLY]. We removed this statement. The sentence now reads:

Page 12, line 28: "Moving eastwards (0°-10° E) elevated dust is trapped topographically by the Alps (47°-52° N) with values >10 $Mm^{-1}$."

**Page 9, line 29-33: You should consider adding here more discussion and some statistical parameters (e.g. mean, standard deviation, maxima, minima, etc) to enrich this summary. Also, some sentences about the dust vertical distribution in the summary are missing.**

[REPLY] Detailed statistics and have been added in our manuscript. The discussion about dust vertical distribution has been also extended:

Page 13, line 14: "In summary, the obtained cross-sections for the five longitudinal zones indicate that higher extinction coefficient values are observed near the source and at low altitudes, where dust particles are efficiently deposited. Above NE Africa, the Clim-DE values are >45 $Mm^{-1}$ throughout the year in altitudes up to 2 $km$ a.s.l. during JFM and up to 4 km during AMJ and JJA. Moreover, the standard deviation of the means is around 130% at the altitudes up to 2 km and ~100% between 2 – 4 km, at all seasons. Above West Africa, the extreme Clim-DE values observed during JAS in the altitudes up to 2 km are 113 ± 131 $Mm^{-1}$. In C-E Mediterranean, dust is always present, with maximum extinctions during AMJ, reaching 27 ± 54 $Mm^{-1}$ close to the surface and ~ 18 ± 30 $Mm^{-1}$ during JAS and OND. In C-W Mediterranean, the highest means of JAS are ~16 ± 40 $Mm^{-1}$. For latitudes greater than 45° N, and during AMJ mean values of 8 ± 27 $Mm^{-1}$ are 4 ± 16 $Mm^{-1}$ are observed close to the surface above NE Europe and NW Europe respectively."

**Page 10, line 1: How is the impact on cloud formation estimated?**

REPLY] The impact of dust on cloud formation is part of a second study we are working on. In this forthcoming work, we will use dust profiles from CALIPSOand EARLINET parameterizations to calculate the dust mass concentration for particles with radius greater than 250 nm and to estimate ice nuclei concentration profilesfollowing the technique provided by Mamouri and Ansmann (2016). We removed this sentence from the revised manuscript to avoid confusion.

**Page 10, line 2: Please, include additional information and discussion on this part related to the dust mass concentration calculation. What is the point of calculating it here?**

[REPLY] This sentenced has been replaced by:

Page 13, line 23: "The dust mass concentration can be obtained from the optical properties of dust with an uncertainty of 20-30% (Ansmann et al., 2012; Mamouri and Ansmann, 2014)."

**Page 10, lines 12-14: The information included here should be provided earlier in the section, before discussing the results.**

[REPLY] The information regarding the Clim-DE and Con-DE products is provided in section 2.3. Here, we repeat the difference between the products in order to introduce the next paragraph, which is devoted to the Con product description. We rephrased the sentence to be clearer:

Page 13, line 28: "The decreasing intensity with height and latitude found in the Clim-DE product is representative of the average dust distribution over the area. However, this behaviour is not representative of the distribution during dust episodes over Europe. This is because the extinction coefficient values presented in Fig. 4 for the Clim-DE product are produced by averaging partially and fully dominated dust cases. In order to describe the spatial patterns and the intensity of the dust plumes during episodes only, we introduce and discuss the Con-DE product in the next section."

**Page 10, line 23: Replace "populations of dust" by "dust features"**

[REPLY] It is replaced.

**Page 10, line 25: Indicate the other seasons and regions where the two distinct layers are observed**

[REPLY] We deleted this part of the paper.

**Page 11, lines 3-12: This paragraph should be moved to later on in the manuscript, in order to keep all the discussion related to figure 4 together. Additionally, more discussion on depolarization should be provided here.**

[REPLY] We deleted this part of the manuscript.

**Page 11, line 16: Replace "in the same range with" by "in the same range as"**

[REPLY] It is replaced.

**Page 11, line 32: At the end of section 3.3 you mentioned that Con-De will be used to discuss if the decreasing intensity with height and latitude is representative, but this is not discussed in section 3.4. Please, include some sentences. Additionally, some more discussion comparing the results from sections 3.4 and 3.3 will be interesting.**

[REPLY] The paragraph at the end of 3.3 has been changed to highlight the difference between the two products, and to justify the need of discussing both.

Regarding the comparison of the two products presented in 3.3 and 3.4, we have included a comment in the first paragraph:

Page 14, line 7: "This is because the two products differ mostly over areas which are not dominated by dust." There is no meaning to our opinion to elaborate further on this comparison, since the difference between the two products has to do with the frequency of occurrence of dust in relation to other aerosol types. Although we introduce a new Table 3 in the end of Sect. 3.4 so as the readers can have a quantitative representation of the two products. The new part is:

Page 15, line 10: "A quantitative representation of the Clim-DE and Con-DE products is provided in Table 3. In this, regional statistics on the two products, along with their standard deviation are provided for three altitudinal ranges ($0-2$, $2-4$ and $4-6$ $km$ a.s.l.)."

Table 3: Regional statistics on the dust extinction coefficient for altitudes between 0 to 2km, 2 to 4 km and 4 to 6 km (a. s. l.).

| | 0 – 2 km | 2 – 4 km | 4 – 6 km | |
|---|---|---|---|---|
| | Clim-DE / Cond-DE / St. dev | Clim-DE / Cond-DE / St. dev | Clim-DE / Cond-DE / St. dev | Domain |
| **NE Africa**
JFM
AMJ
JAS
OND | $42 / 50 / 74$ $Mm^{-1}$
$66 / 66 / 88$
$42 / 42 / 64$
$34 / 34 / 51$ | $7 / 43 / 20$ $Mm^{-1}$
$44 / 53 / 48$
$30 / 40 / 37$
$17 / 32 / 24$ | $0 / 25 / 5$ $Mm^{-1}$
$18 / 48 / 26$
$13 / 43 / 22$
$3 / 27 / 9$ | [10E,30E]
[20N,30N] |
| **NW Africa**
JFM
AMJ
JAS
OND | $46 / 60 / 80$ $Mm^{-1}$
$73 / 73 / 90$
$113 / 113 / 131$
$59 / 59 / 86$ | $6 / 45 / 18$ $Mm^{-1}$
$41 / 59 / 49$
$83 / 83 / 71$
$35 / 48 / 43$ | $0 / 29 / 5$ $Mm^{-1}$
$13 / 51 / 25$
$43 / 50 / 40$
$10 / 36 / 19$ | [10W,10E]
[20N,35N] |
| **C-E Med.**
JFM
AMJ
JAS
OND | $22 / 44 / 55$ $Mm^{-1}$
$27 / 35 / 54$
$18 / 18 / 28$
$19 / 23 / 32$ | $4 / 48 / 16$ $Mm^{-1}$
$17 / 52 / 34$
$13 / 33 / 22$
$10 / 35 / 19$ | $0 / 31 / 5$ $Mm^{-1}$
$5 / 42 / 15$
$4 / 37 / 12$
$2 / 27 / 7$ | [10E,30E]
[30N,45N] |
| **C-W Med.**
JFM
AMJ
JAS
OND | $5 / 24 / 33$ $Mm^{-1}$
$10 / 23 / 38$
$16 / 22 / 40$
$10 / 22 / 33$ | $1 / 32 / 7$ $Mm^{-1}$
$6 / 35 / 19$
$13 / 33 / 23$
$4 / 29 / 14$ | $0 / 21 / 2$ $Mm^{-1}$
$1 / 31 / 8$
$5 / 38 / 14$
$0 / 29 / 4$ | [10W,10E]
[35N,45N] |
| **NE Europe**
JFM
AMJ
JAS
OND | $4 / 37 / 41$ $Mm^{-1}$
$8 / 17 / 27$
$7 / 14 / 21$
$4 / 16 / 19$ | $0 / 29 / 5$ $Mm^{-1}$
$2 / 21 / 17$
$2 / 16 / 9$
$1 / 21 / 6$ | $0 / 15 / 1$ $Mm^{-1}$
$0 / 14 / 2$
$0 / 16 / 2$
$0 / 14 / 1$ | [10E,30E]
[45N,60N] |
| **NW Europe**
JFM
AMJ
JAS
OND | $1 / 16 / 16$ $Mm^{-1}$
$4 / 16 / 16$
$3 / 15 / 15$
$2 / 16 / 15$ | $0 / 16 / 2$ $Mm^{-1}$
$1 / 21 / 11$
$1 / 22 / 7$
$0 / 23 / 4$ | $0 / 15 / 1$ $Mm^{-1}$
$0 / 14 / 2$
$0 / 18 / 2$
$0 / 13 / 0$ | [10W,10E]
[45N,60N] |

**Page 12, line 18: Replace "statistical significant" by "statistically significant"**

[REPLY] It is replaced.

---

## Author Comment (AC2) · 3 Mar 2017

**General remarks:**

**The present manuscript provides a monthly climatology (from 2007 to 2015) of African dust based on an optimised CALIPSO dust product was recently developed with a regional correction of the Saharan dust LR using EARLINET measurements (Amiridis et al., 2013). The monthly climatology of African dust obtained allows the description of the spatiotemporal features of dust properties over North Africa and Europe. The study of the mean state climatology shows strong seasonal shifts in dust source regions and transportation pathways. While the results of the study are interesting to be published, their presentation and discussion are not yet sufficient to be published in Atmospheric Chemistry and Physics in the current form. Therefore, it is worth to be published after addressing major revisions which are explained below along with a few other details.**

[REPLY] We thank the reviewer for the thorough revision and comments. Replies to the general and specific comments follow below.

**Major comments:**

**In Amiridis et al. (2013), this EARLINET-optimized CALIPSO dust optical depth (for the period 2007-2011) is described and qualitatively compared with MODIS and AERONET. The present manuscript is focusing on the analysis of the resulting EARLINET-optimized CALIPSO dust climatology. I would be desirable to include a short discussion of the uncertainties of the EARLINET-optimized CALIPSO dust product. I understand that this discussion is partly in Amiridis et al. (2013, 2015) although the authors should include a summary in Sect. 2.2 as well as about the uncertainties of the algorithm of CALIOP to determine the corresponding aerosol subtype (in Sect 2.1).**

[REPLY] We added a new section 3.5 discussing the uncertainties related to the EARLINET-optimized CALIPSO dust product. Moreover, we added a summary in Sect. 2.2 about the uncertainties of the algorithm of CALIOP to determine the corresponding aerosol subtype, referring to the evaluation done with NASA's HSRL and presented in Burton et al. (2013).

[revised manuscript text omitted]

**In Figure 1, there are some features that they look associated with the number of available observations, and consequently with the presence of clouds over the Mediterranean and Europe. I am not sure if the "%Dust/Used Overpasses" is enough to explain the DOD seasonal patterns in Europe. I would suggest to include an additional column with the number of used overpasses and to check how is working the algorithm of CALIOP to determine the corresponding aerosol subtype in this part of the domain.**

[REPLY] The capability of CALIPSO to detect the dust subtype has been thoroughly evaluated by Burton et al., (2013) using a number of 109 underflights of CALIPSO with NASA's HSRL system and found that the detection of dust from CALIPSO is successful in 80% of the compared cases. Figure 1 is meant to present the frequency of CALIPSO dust occurences in the domain of our interest, in relation to the cloud-free overpasses of the sensor over the area. Following the helpful comment of the reviewer a new Table was added in the manuscript (Table 3), in order to provide a more informative representation of the dataset, including the percentages of the cloud free observations used, in relation to the total observations provided, aggregated on 6 areas over the study region. Furthermore we added the following discussion in the manuscript (Page 9, line 14):

Table 2 shows the impact of cloud contamination in our dataset. During AMJ, JAS and OND, more than 80% of the total observations are cloud-free above North Africa. Above Central-East Mediterranean (C-E Med.), more than 80% of the total observations are cloud-free and above Central West Mediterranean (C-W Med.) approximately 60% - 80% of the total observations are cloud-free. With increasing latitude, the cloud-free sampling is reduced to percentages of ~ 40% -60% in latitudes greater than 45° N. During JFM, cloudy conditions restrict our dataset in the greatest extent. During the same period, the cloud-free cases used represent ~ 80% of the total observations above North Africa, approximately 60 - 70% of the total observations above the Mediterranean and ~ 30% in the domain between 45° N - 60° N.

In the areas (and seasons) where clouds do not dominate (e.g. 70% clear-sky conditions), our cloud-free product is considered representative of the dust distribution. In areas where cloudy skies dominate (e.g. 30% clear-sky conditions), the clear-sky CALIPSO profiles cannot be considered as representative of all meteorological conditions, so the results should be used with caution."

**In Page 10 Line 1, you mention that the results from Clim-DE can be used to estimate the impact of dust on cloud formation. As far as I understood, the EARLINEToptimized CALIPSO dust product is provided only for clear-sky conditions. In Sect. 4, you mention a recent paper from Mamouri and Ansmann (2016), but it is based on a ground-based lidar. Then, how could you estimate the dust impact on cloud formation from this EARLINET-optimized CALIPSO dust product?**

[REPLY] The impact of dust on cloud formation is part of a second study we are working on. In this work, we use dust profiles from CALIPSO, in combination with EARLINET parameterizations, in order to calculate the dust mass concentration for particles with radius greater than 250nm and from there, based on known ice nuclei parameterizations to estimate ice nuclei concentration profiles. A detailed analysis of this technique is provided in the work of Mamouri and Ansmann (2016). An example of the application of this technique on collocated ground-based and CALIPSO data and comparison with in situ estimated ice nuclei will be presented in the upcoming ILRC Conference in Budapest. (Marinou, et al.: Lidar ice nuclei estimates and how they relate with airborne in-situ measurements, 28th ILRC, Bucharest, 25-30 July 2017). In order to keep the discussion as straightforward as possible, and to avoid confusing the readers, we decided to delete the corresponding part.

**In my opinion, a further discussion about the similarities and discrepancies with other dust climatologies will enhance the impact of the results presented in the manuscript. Any comparison with other dust climatologies based on other datasets such as satellites (e.g. MODIS, AERONET, EARLINET or the official CALIPSO aerosol product); and models (as CAMS reanalysis or AEROCOM) is considered in the manuscript. Furthermore, how do the results of the present study improve those results of LIVAS (Amiridis et al., 2015)? These discussions will be useful for model evaluation, for example. Otherwise, it seems to me that some results are general and not enough justify in the manuscript.**

[REPLY] We thank the reviewer for this comment. Amiridis et al., (2015), did not analyze the dust transport patterns. That paper describes the LIVAS database and focuses on the spectral conversion of the CALIPSO 532nm products for use in future ESA lidar missions that operate at 355nm. Following the suggestion of the reviewer, we calculated the optical depth using other available products such as the AOD from MODIS data and the DOD from MACC and RegCM4 models. We added a new figure, Figure 2, comparing our DOD seasonal maps with the ones produced with the above mentioned products, that is followed by a short discussion (page 10, line 26):

"In order to provide a more informative representation of the dust product presented here, we performed a comparison with MODIS AOD for the same period and the dust optical depth of the MACC reanalysis and a RegCM4 simulation for the period 2007-2012 and 2007-2014 respectively (Fig. 2). MODIS provides AOD for all natural and anthropogenic aerosol types. As

a result the MODIS average value for the whole period and domain (0.267) is 281% almost three times, bigger than our product (0.095). It is noted thought that the values between the two satellite products are very similar over the Sahara desert. On the contrary, the corresponding average dust optical depth values of MACC (0.100) and RegCM4 simulations (0.104) reproduce better our product, since only dust is considered, though our product is lower by 5% and 8.6% respectively. Dust optical depth is overestimated over Europe and Mediterranean by MACC and RegCM4 simulations in comparison to our product in all seasons and especially in the hot periods AMJ and JJA, but the reasons of these discrepancies have to be further studied."

**In Sect. 3.1 (Page 10) I don't understand the reason to include the dust mass concentration inversion results. This part of the discussion doesn't include any new insight with respect the analysis of the optical properties or any link to a particular previous study.**

[REPLY] We removed the formulas and relevant discussion to the dust mass concentration inversion from the paper.

**In Sect. 3.5, you could compare your results with a climatic index as the North Atlantic Oscillation Index (NAO) as Pey et at. (2013) did for PM10.**

[REPLY] Following the analysis of Moulin et al. (1997) and Pey et al. (2013) we investigated the relation between NAO index and LIVAS AOD in seasonal and monthly basis (i.e., summer, winter, annually) for the period of our study but  we did not find a statistically significant correlation and thus we did not include it in our results. Especially for summer the correlation between the NAO index and the LIVAS DOD over the western Mediterranean is negative but not statistically significant.

**Is this de-seasonalised trend analysis sensitive to the number of available observations?**

[REPLY] In our study, we calculate the DOD trend along with the statistical significance of each trend for the period 2007-2015 (108 monthly values). Nine years are considered a small period for a robust trend calculation and it would be interesting to repeat the same analysis in the future to an extended aerosol record. The de-seasonalization process as well as the trend are described only for the examined period only. In case we extend our analysis in the future by adding more years, results may change. We added this clarification in the manuscript (page 16, line 13):

"In our study, we calculate the DOD trend along with the statistically significance of each trend for the period 2007-2015 (108 monthly values). Nine years are considered a small period for a robust trend calculation and it would be interesting to repeat the same analysis in the future to extended aerosol record. The de-seasonalization process as well as the trend are describing the examined period only. Figure 7 shows the DOD internal variability of the 20 individual areas, as it is calculated from monthly mean DODs. Is evident from this figure that the DOD values in 2008 are relatively higher than the other years and in almost all the domains bellow 40°N. Similarly, relatively high values are observed in some of these areas for the year 2010. Since these years are at the beginning of our study period, they have a significant contribution on the negative trends observed during the examined period."

**Minor comments:**

**Page 2 Line 13: Add Nickovic et al. (2016).**

[REPLY] We added this reference.

**Page 2 Line 16: Add Granados-Muñoz et al. (2016) and Bovchaliuk et al. (2016).**

[REPLY] We added these references.

**Page 3 Line 25: Replace Gkikas et al. (2015, ACPD) by Gkikas et al. (2016, ACP).**

[REPLY] It is replaced.

**Page 3 Line 30: When you said "large scale statistics of discriminate and optimized dust extinction and AOD fields from CALIPSO", what does it mean? What about Amiridis et al. (2013) and Amiridis et al. (2015)?**

[REPLY] This phrase is removed and replaced by the phrase (page 4, line 4):

 "To our knowledge, this is the first time that a 3D pure-dust dataset is statistically analysed over the area of North Africa and Europe in order to provide not only the horizontal but also the vertical patterns of Saharan dust intrusion in the Mediterranean."

Amiridis et al., (2013 and 2015) did not analyze the dust transport patterns. Amiridis et al (2013) describe the methodology for the pure-dust retrieval algorithm. Amiridis et al., (2015) describes the LIVAS database. The later paper focuses on the spectral conversion of the CALIPSO 532nm products for use in future ESA lidar missions that operate at 355 nm.

**Sect. 3.1: It would be good if you can add a short comparison of the resulting DOD seasonal maps with the results of MODIS, MISR or any available reanalysis (as CAMS or MERRA).**

[REPLY] We thank the reviewer for this comment. Therefore we calculated the optical depth using other available products such as the AOD from MODIS data and the DOD from MACC and RegCM4 models. We added a new figure, Figure 2, comparing our DOD seasonal maps with the ones produced with the above mentioned products, that is followed by a short discussion (page 10, line 26):

 "In order to provide a more informative representation of the dust product presented here, we performed a comparison with MODIS AOD for the same period and the dust optical depth of the MACC reanalysis and a RegCM4 simulation for the period 2007-2012 and 2007-2014 respectively (Fig. 2). MODIS provides AOD for all natural and anthropogenic aerosol types. As a result the MODIS average value for the whole period and domain (0.267) is 281% almost three times, bigger than our product (0.095). It is noted thought that the values between the two satellite products are very similar over the Sahara desert. On the contrary, the corresponding average dust optical depth values of MACC (0.100) and RegCM4 simulations (0.104) reproduce better our product, since only dust is considered, though our product is lower by 5% and 8.6% respectively. Dust optical depth is overestimated over Europe and

Mediterranean by MACC and RegCM4 simulations in comparison to our product in all seasons and especially in the hot periods AMJ and JJA, but the reasons of these discrepancies have to be further studied."

**Figure 2: Comparison of the seasonal spatial distribution of the optical depth as received by (first column) pure-dust CALIPSO DOD product, (second column) MODIS AOD product, (third column) MACC reanalysis DOD product, (fourth column) RegCM4 simulated DOD product.**

[Figure]

**Sect. 3.2: In this case, you could compare your results from those obtained from EARLINET or models.**

[REPLY] We thank the reviewer for this suggestion. We added the following paragraph in section 3.2 (page 11, line 29):

"In general, our results are in agreement with lidar-based studies which have been performed in several European sites. Papayannis et al. (2008) performed an exhaustive analysis on Saharan dust particles over Europe using EARLINET lidar profiles. They found that the dust layer center of mass extends from 3.0 to 3.8 km and the thickness ranges from 0.7 to 3.4 km. Specifically, Balis et al. (2012) calculated the mean base and top of dust layers in the eastern Mediterranean, Thessaloniki, to be around $2.5 \pm 0.9\ km$ and $4.2 \pm 1.5\ km$, respectively. More

recently, Mona et al. (2014) analyzed a long dataset of Saharan dust intrusions over Potenza, Italy, and found a mean layer centre of mass of $3.5 \pm 1.5\ km$."

**Sect 3.5: how do your results fit with those showed in Gkikas et al. (2016)?**

[REPLY] In Gkikas et al. (2016) there is no discussion on the interannual variability of the dust events. The interannual variability of dust events is discussed in Gkikas et al. (2013). A sentence is added in the manuscript comparing our results to this work (page 15, line 32):

"Furthermore, over the same domain the decreasing trend of DOD coincides with the decrease of Saharan desert dust episodes as reported by Gkikas et al. (2013)."

**Page 10 Line 28: Add Huneeus et al. (2016).**

[REPLY] We added this reference.

**Page 10 Line 30: "it is likely that the surface and elevated dust have different origins" sounds speculative. You could check this assumption with models or back trajectories.**

[REPLY] The corresponding sentence is removed from the revised manuscript.

**Figure 3. I would use the same colour palette than in Figure 4.**

[REPLY] Figures 3 and 4 (new Figures 4 and 5) are based on the same color pallet with the difference that in Figure 4 a new restriction is introduced. In both cases, the black values represent the mean terrain elevation. In Figure 5, when we average the Con-DE, there is different sampling than the one used for the Clim-DE, and as a result, some means are produced from very few numbers of dusty observations (dO). In order to filter these case from the plot, so as the readers to concentrate on more significant features, we mask them with the new gray color. In order to better address these filters in the plots we changed the color pallet of Figure 5, and added a NaN black box (similar as in Figure 4), and labeled the gray box as "<4dO".

We also changed the manuscript when introducing the two figures:

In section 3.3 (page 12, line 11): "The median surface elevation is depicted with black colour (and is labeled as NaN) in the plots.

In section 3.4 (page 14, line 3): "Con-DE values derived from less than 4 dust observations (dO) in each cell are masked with grey colour (and are labeled as <4dO) in the plots. The median surface elevation is depicted with black colour (same as in Fig. 4)."

**Figure 3. Could you provide any further explanation about the sharp transition over the Atlas?**

[REPLY] We added the following sentence in the paper (page 12, line 32):

"The area south of Atlas Mountains (Fig. 4e, f, g, h) is characterized by haboob activity (Knippertz et al., 2009; Solomos et al., 2012). These systems are generated from convective outflows and contribute to the interannual burden of dust at this area."

**Figures 3,4,5: Longitude and Latitude labels can be removed. They are too small.**

[REPLY] The labels size is increased and it is more visible in the new version of the manuscript.

---

## Author Comment (AC3) · 4 Mar 2017

**In this manuscript, "3D evolution of Saharan dust transport towards Europe based on a 9-year EARLINET-optimized CALIPSO dataset", the authors use a combination of CALIPSO and EARLINET to present a climatology of recent dust vertical distribution and transport to Europe from Africa. The consideration of both climatological extinction and 'conditional' extinction is useful to elucidate the episodic transport as well as the dust distribution.**

**The manuscript is generally well written and the climatology will be useful to the community and to evaluate models. However, there is a lack of discussion of uncertainties in the product and some limited interpretation of the particle depolarization ratio and interannual variability that need revision. Please see the major and minor comments below.**

[REPLY] We thank the reviewer for the thorough revision and comments. We agree with the importance of the discussion on the uncertainties in the product, thus we added a new section (Section 2.4) discussing all the uncertainties of the product in the manuscript. We decided to delete the particle depolarization ratio discussion from the paper, so as to help the reader concentrate on the other parts of this work. The section of the interannual variability is substantially revised. Replies to general and specific comments can be found below.

**Major Comments**

**There is limited discussion of the uncertainties and detection limits of dust occurrences throughout the manuscript. There are some very high occurrences of dust shown in regions far from dust sources in Figure 1 (e.g. the North Atlantic). How certain are we that this is actually dust and based on what detection limit? Similarly, Figures 3 and 4 show infrequent but high extinction dust at the surface at high latitudes. Can we besure this is not a retrieval artifact? When climatological extinctions as low as 5 Mm-1 are considered (e.g. pg9 line 22) it would be useful to know the uncertainty on the estimates.**

[REPLY] We thank the reviewer for these comments. We added a new section (Section 2.4) discussing the uncertainties of the produced product. In the same section, we mention also the detection limit for dust occurrences and the uncertainty introduced from this choice by stating: "Moreover, we have calculated that the uncertainty of the dust occurrences presented in Sec. 3.1 ("% Dust / Used Overpasses"), might be up to 8% in latitudes away from the sources, induced from the error in the selection of the $\delta_{nd}$ value (0.03±0.04)." In a more detail explanation on how this percentage is estimated: the selected detection limit is based on depolarization measurements, and any layer with depolarization values greater than 0.03 is considered as mixture of dust with other aerosols. This detection threshold correspond to the lowest the depolarization values found in nature for clean marine, smoke and anthropogenic aerosols, i.e., $0.03 \pm 0.01$, $0.06 \pm 0.01$ and $0.06 \pm 0.01$, respectively. We estimated the uncertainty that this detection limit may induce in the occurrences of dust far from sources. For cases where the depolarization of the non-dust feature is $0.03 < \delta_{nd} \leq 0.075$, the low selected $\delta_{nd}$ value, may introduce error as high as 100%. In CALIPSO dataset of our domain, these cases correspond to less than 4% of the dust and polluted dust layers

used (1% of the dust layers used and 8% of the polluted dust layers used). This uncertainty is transferred to the uncertainty of the dust occurrences presented in Sec. 3.1, inducing a positive bias up to 8% in latitudes away from the sources for the parameter "% Dust / Used Overpasses", as this parameter refers to observations with DOD > 0.

In Figures 3 and 4 (new figures 4 and 5), the uncertainty of the dust extinction values close to the surface and at high latitudes are <54%. At high altitudes and for latitudes up to 45°N, the uncertainty of the values in these figures is <20%. Nevertheless, the standard deviation of the Cilm-DE product, originating from the natural variability of the dust events, may excess to a large extent the uncertainty of the retrieval, reaching values from 100% to 200%.

The following section is added in the manuscript (page 7, line 31):

**"2.4 Dust product uncertainties**

The sources of uncertainties for the pure-dust product are discussed in this section. CALIOP is able to detect aerosol layers with $AOD > 0.005$ and $\beta > 0.25\ Mm^{-1}\ sr^{-1}$ (Winker et al. 2009). The uncertainty estimation of particulate backscatter, extinction and AOD retrievals reported in the CALIPSO Level 2, Version 3 Data Release, are based on the simplified assumption that all the uncertainties are random, uncorrelated and produced no biases (Young, 2010). More specifically, ignoring multiple scattering, the errors in the layer optical depth calculations typically arise from three main sources: (a) signal-to-noise ratio within a layer, (b) calibration accuracy, and (c) the accuracy of the lidar ratio used for the extinction retrieval. The lidar ratio uncertainty is the dominant contributor to the total uncertainties, and the relative error in the layer optical depth is always at least as large as the relative error in the lidar ratio of the layer, and grows as the solution propagates through the layer (CALIPSO L2-V3, 2010). In our dataset the typical uncertainties in the CALIPSO Level 2 version 3 product are between 30% and 100% for the AOD, between 30% and 160% for the aerosol backscatter and extinction coefficient and >100% for the particle depolarization ratio.

Several studies report that CALIPSO underestimates the columnar AOD due to undetected aerosol in the free atmosphere. For instance, Rogers et al. (2014) report a ~0.02 AOD CALIPSO underestimation, when compared to collocated airborne HSRL measurements over the North American and Caribbean regions at night. In their data, the dust layers were primarily non-opaque with extinction less than $1\ km^{-1}$ so there were negligible multiple scattering effects. The aforementioned detection limits and uncertainties of CALIPSO products are propagated to the dust product presented here.

As already described, the EARLINET-optimized CALIPSO dust product is derived using the depolarization-based separation method, coupled with the selection of a uniform climatological LR value. These steps introduce uncertainties in the pure dust product. In particular, the uncertainty in the selection of the representative LR ($55 \pm 11$) is 20% for the study area (e.g. Wandinger et al. 2010; Baars et al. 2016 and references within). This uncertainty in LR is less than half of the uncertainty of the generic LR in CALIPSO version 3 product ($40 \pm 20$ for dust layers and $55 \pm 22$ for polluted dust layers). As already addressed in several studies (e.g. Wandinger et al. 2010; Schuster et al. 2012; Amiridis et al. 2013), CALIPSO V3 dust extinction coefficient and AOD values are about 30% lower than those

obtained from collocated ground-based Raman lidar retrievals due to the low LR used in the CALIPSO aerosol retrievals. Amiridis et al. (2013) applied the EARLINET LR for the pure dust CALIPSO cases above North Africa and Europe, and compared with synchronous and collocated AERONET measurements. The results showed an absolute bias on the AOD of the order of −0.03, improving on the statistically significant biases of the order of −0.10 reported in the literature for the original CALIPSO product. The bias of -0.03 is similar to the low bias of CALIPSO's column AOD due to undetected aerosol layers. In Kim et al. (2017), they found a global mean undetected layer AOD of $0.0031 \pm 0.052$ by comparing 2 year of CALIPSO (L1-V4) and MODIS AODs.

Regarding the error induced from the application of the dust separation method, this might be due to the selection of the particle depolarization ratio of dust and the other aerosol types (marine, anthropogenic or smoke). Tesche et al. (2009; 2011) and Ansmann et al. (2012) estimated that the uncertainty in dust related backscatter coefficients is 15-20% in well-detected dessert dust layers and 20-30% in less pronounce aerosol layers. Moreover, we have calculated that the uncertainty of the dust occurrences presented in Sec. 3.1 ("% Dust / Used Overpasses"), might be up to 8% in latitudes away from the sources, induced from the error in the selection of the $\delta_{nd}$ value (0.03±0.04). Finally, an uncertainty induced in the dust product presented in this work, originates from the CALIPSO subtype selection algorithm. In this version of our product, both dust and polluted dust observations are considered polluted dust, and the pure dust component is separated using the dust separation method. The other aerosol layers, which are characterised as clean marine (CM), smoke (S), polluted continental (PC) or clean continental (CC) are considered to be cases clear of dust and are not tested for a dust component. This introduces negligible error in our analysis and is expected to induce a negative bias in the parameter "% Dust / Used Overpasses" less than 8%, mainly in areas above sea. In general, Clim-DE and Cond-DE products, the uncertainty of the dust extinction values close to the surface and at high latitudes is < 54%. At high altitudes and for latitudes up to 45°N, the uncertainty of the values is < 20%. Nevertheless, the standard deviation of the climatological products, coming from the natural variability of the dust events, may exceed to a large extent the uncertainty of the retrieval, reaching values as high as 100% and 200%.

In the latest release of CALIPSO Level 2 version 4 product (CALIPSO L2-V4, 2016), based on CALIPSO team announcement, the accuracy of the original CALIPSO product is increased and the uncertainty is reduced. This version is based on a revised calibration approach which leads to an increase in the total attenuated backscatter coefficients by ~3% overall as compared to the version 3 values (CALIPSO L1-V4, 2016). Several bugs are fixed and a major overhaul of the aerosol subtyping algorithms along with revisions on the lidar ratio selections is applied."

**The retrieval is provided only for clear-sky conditions. Can you comment on how this might bias the dust extinction and how it relates to cloud formation (mentioned on page 10)?**

[REPLY] We thank the reviewer for this comment. Indeed the restrictions of the dataset, related to the cloudy meteorological conditions were not addressed in the manuscript, therefore we commented analogously below.

First, let us address the second part of the question, to comment on how it relates to cloud formation (mention on page 10). The impact of dust on cloud formation is part of a second

study we are working on. In this work, we use dust profiles from CALIPSO, in combination with EARLINET parameterizations, in order to calculate the dust mass concentration for particles with radius greater than 250 nm and from there, based on known ice nuclei parameterizations to estimate ice nuclei concentration profiles. A detailed analysis of this technique is provided in the work of Mamouri and Ansmann (2016). In order not to confuse the readers, we decided to delete the part where we mention "and the impact on cloud formation" from our manuscript.

Regarding the first part of the reviewer's comment, we added a new Table in the manuscript, in order to provide a more informative representation of the dataset. In this table (Table 3) the percentages of the cloud free observations used, in relation to the total observations are provided, aggregated on 6 areas over the study region. Furthermore we added the following discussion in the manuscript (page 9, line 14):

"Table 2 shows the impact of cloud contamination in our dataset. During AMJ, JAS and OND, more than 80% of the total observations are cloud-free above North Africa. Above Central-East Mediterranean (C-E Med.), more than 80% of the total observations are cloud-free and above Central West Mediterranean (C-W Med.) approximately 60% - 80% of the total observations are cloud-free. With increasing latitude, the cloud-free sampling is reduced to percentages of ~ 40% - 60% in latitudes greater than 45° N. During JFM, cloudy conditions restrict our dataset in the greatest extent. During the same period, the cloud-free cases used represent ~ 80% of the total observations above North Africa, approximately 60 - 70% of the total observations above the Mediterranean and  ~ 30% in the domain between 45° N - 60° N. In the areas (and seasons) where clouds do not dominate (e.g. 70% clear-sky conditions), our cloud-free product is considered representative of the dust distribution. In areas where cloudy skies dominate (e.g. 30% clear-sky conditions), the clear-sky CALIPSO profiles cannot be considered as representative of all meteorological conditions, so the results should be used with caution."

**In Figure 1 there is a strong boundary along the European coastline for dust occurrences, is this the result of a marked difference in used overpasses between the mainland and the Mediterranean? Please make sure that this feature is explained.**

[REPLY] Unfortunately, we cannot see the boundary the reviewer is referring to along the European coastlines. There might be a boundary along the French coastlines, however it looks like that over the Iberian Peninsula, Greece and Italy the number of dust overpasses well penetrate over land.

**I don't think simply listing papers that have used specific instruments to explore dust over the Mediterranean is the best way of presenting the introduction (pg3 lines 10-20). Please consider reconstructing this paragraph to briefly discuss what these papers show that is relevant to understanding dust transport to Europe, rather than framing around the instrument used.**

[REPLY] We reconstructed this paragraph according to the reviewer's suggestions. The new paragraph is (page 3, line 8):

"Many studies have used satellite observations to derive dust properties over the Mediterranean during the last 15 years. Most of them focus on the horizontal distribution of dust using passive remote sensing techniques. Antoine and Nobileau, (2006) used SeaWIFS (Sea-Viewing Wide Field-of-View Sensor) observations to study the seasonal evolution and variability of dust aerosols over the broader Mediterranean Sea during the period 1998-2004. Alpert and Ganor (2001) and Israelevich et al. (2002) used the Total Ozone Mapping Spectrometer (TOMS) Aerosol Index (AI) product in order to study the concentration of dust over Middle East and the dust sources of Northern Africa, respectively. The MODIS instrument, onboard both Terra and Aqua satellites has been extensively used in studies of airborne mineral dust over the Mediterranean basin. Barnaba and Gobbi (2004) analysed one-year (2001) MODIS/Terra AOD at 550 nm observations and reported on the spatial distribution and seasonal variability of aerosols, including dust, over the Southern Europe, with a focus over the Mediterranean region. Papayannis et al. (2005) used MODIS/Terra data synergistically with lidar measurements and dust model simulations and investigated the vertical distribution of aerosols during dust outbreaks over Greece. Kosmopoulos et al. (2008) and Papadimas et al. (2008) used MODIS/Terra and MODIS/Aqua to investigate the seasonal and interannual variability of AOD at 550 nm over Athens (Greece) and over the broader Mediterranean Sea, respectively. Marey et al. (2011) analysed ten-years of MODIS data synergistically with MISR and OMI and they produced a monthly climatology of aerosols over a domain covering the Nile Delta and northeast Africa."

**The seasonal climatological and conditional meridional dust extinction product will be useful for evaluating model representation of dust transport to Europe. I recommend that the authors make this available to the research community and include a link to the dataset in the manuscript, if possible.**

[REPLY] We added a new section for data availability where we provide the availability of this dataset. In the new Section 5 we provide this information (page 17, line 22) :

"The LIVAS database is publicly available at http://lidar.space.noa.gr:8080/livas/. LIVAS EARLINET-optimized pure dust products are available upon request from Eleni Marinou (elmarinou@noa.gr) and Vasilis Amiridis (vamoir@noa.gr)."

**The section on interannual variability is quite weak. The comparisons with other studies should be relevant to the time period considered in this work.**

[REPLY] We agree with the reviewer that a more extended discussion on the interannual variability section would improve the structure of the manuscript. The referenced literature was mainly focused on studies related to the decrease of both dust concentration and frequency close to the surface, for two basic reasons. Firstly, most of the available studies focus on the interannual variability and trends of AOD, not of DOD. This is due to the difficulty of disentangling the dust component of the total aerosol load. Secondly, interannual variability studies have been carried out mainly over the second half of the past decade and mostly by using columnar SeaWiFS, MODIS (Aqua/Terra), MISR, AVHRR and AERONET data. In this study, CALIIOP/CALIPSO vertically-resolved observations in nine years are used, which provide an accurate and robust way of identifying mineral dust from space. Furthermore, the methodology has been established and validated based on EARLINET for CALIPSO mineral dust

research. In this way, CALIPSO is considered as an ideal tool from space to decouple the dust component from the total aerosol burden and for studies of the variability of DOD. Nevertheless, since the authors agree with the reviewer and in order to ratify the results, modifications on the manuscript were made. The authors extended the list of referenced studies related to the interannual variability not only for DOD but additionally for AOD, with an effort to be more focused over the study region. The "Interannual variability of dust" section is modified by adding the following text (page 15, line 29):

"In comparison with studies relevant to the time period considered in this work, the DOD decrease of 0.001 $yr^{-1}$ over the northern coast of Africa is in agreement with Floutsi et al. (2016), who based on 12 years of MODIS-Aqua observations (2002-2014) reported an average decrease of 0.003 $yr^{-1}$ for the coarse mode fraction of AOD over the broader Mediterranean Sea. Furthermore, over the same domain the decreasing trend of DOD coincides with the decrease of Saharan desert dust episodes as reported by Gkikas et al. (2013). Regarding the AERONET stations over the domain of northern Africa and Europe, Yoon et al. (2012) reported on the trends of AOD at 440 nm along with the corresponding Ångström Exponents (440 and 870nm). The documented negative trends over the AERONET stations of Avignon (France), Dakar (Senegal) and Ispra (Italy) are in agreement with the negative DOD reported here, although with discrepancies in the magnitude, while trend disagreements are observed over the AERONET station of Banizoumbou (Niger). The decreasing trends of DOD observed over the domain northern of Africa and Europe coincide with the generally documented downward AOD trends reported based on several satellite observations of MODIS/Aqua, MODIS/Terra, MISR and SeaWiFS (Pozzer et al., 2015; de Meij et al., 2012; Hsu et al., 2012; Georgoulias et al. 2016b). More particular, in the most recent study of Georgoulias et al. (2016b), using MODIS/Terra and MODIS/Aqua observations, they reported negative statistically significant trends over Algeria, Egypt and the Mediterranean and positive trends over Middle East. Overall, for the Mediterranean they reported an AOD trend of -0.0008 $yr^{-1}$ for the MODIS/Terra observations (2000 – 2015) and -0.0020 $yr^{-1}$for the MODIS/Aqua observations (2002 – 2015), with the trends being statistical significant at the 95% confidence level in both cases."

**There does seem to be a general downward trend, but based on the lack of significance in many regions it is understandably difficult to determine long term trends over a relatively short 8 year period. Maybe the authors could include a timeseries panel in Figure 6 to indicate the interannual variability, rather than focus on the weak trends?**

[REPLY] In our study, we calculate the DOD trend along with the statistical significance of each trend for the period 2007-2015 (108 monthly values). We also believe that trends might change if a longer time scale is used. According to the reviewer's suggestion, we added a timeseries panel as Figure 7, including the interannual variability of the 9 year observations based on monthly mean DODs. The interannual section was modified, and the following text is added (page 16, line 13):

"In our study, we calculate the DOD trend along with the statistically significance of each trend for the period 2007-2015 (108 monthly values). Nine years are considered a small period for a robust trend calculation and it would be interesting to repeat the same analysis in the future

to extended aerosol record. The de-seasonalization process as well as the trend are describing the examined period only. Figure 7 shows the DOD internal variability of the 20 individual areas, as it is calculated from monthly mean DODs. Is evident from this figure that the DOD values in 2008 are relatively higher than the other years and in almost all the domains bellow 40°N. Similarly, relatively high values are observed in some of these areas for the year 2010. Since these years are at the beginning of our study period, they have a significant contribution on the negative trends observed during the examined period."

**Figure 7: Interannual variability of the DODs for the 10° x 10° grid cells depicted in Fig. 6, for the period 2007-2015.**

[Figure]

**Minor Comments**

**Please replace all instances of 'utilize' with 'use'**

[REPLY] It is replaced  throughout the manuscript

**pg1 ln21 - "During spring..." sentence is not clear, please revise.**

[REPLY] We revised the sentence as (page 1, line 21): "During spring, the spatial distribution of dust shows a uniform pattern over the Sahara desert."

**pg1 ln22 - "on" should be "in"**

[REPLY] It is replaced.

**pg1 ln23 - "0.1", should this be "up to 0.1"?**

[REPLY] It is rephrased: "The dust transport over the Mediterranean Sea results in mean Dust Optical Depth (DOD) values up to 0.1."

**pg1 ln28 - units are sometimes italicized, other times not**

[REPLY] We harmonized the units format in the manuscript, and, now, they are italicized everywhere.

**pg1 ln31 - change to "the Alps and Carpathian Mountains"**

[REPLY] It is changed.

**pg2 ln25 - remove "now"**

[REPLY] It is removed.

**pg3 ln30 - what is meant by "large scale statistics"**

[REPLY] This phrase is removed and replaced by the phrase (page 4, line 4):

"To our knowledge, this is the first time that a 3D pure-dust dataset is statistically analyzed over the area of North Africa and Europe in order to provide not only the horizontal but also the vertical patterns of Saharan dust intrusion in the Mediterranean."

**pg4 ln22 - extra space after "biases"**

[REPLY] It is removed.

**pg5 ln13 - perhaps provide the link as a reference?**

[REPLY] We provide the link as a reference now. In the discussion (page 5, line 20):

"In brief, CALIPSO L3 version 3 screening procedure is followed (Winker et al., 2013; CALIPSO L3-V3, 2015)"

Reference:

CALIPSO L3-V3: CALIPSO: Data User's Guide - Data Quality Statement - Lidar Level 3 Aerosol Profile Monthly Product Version 3.00, link: http://www-calipso.larc.nasa.gov/resources/calipso_users_guide/data_summaries/l3/CALIOP_L3Products_3-00_v01.php, 2015.

**pg5 ln21 - "categorized"**

[REPLY] It is removed.

**pg6 ln17 - "However..." it is not clear why this is an issue. Please elaborate or remove.**

[REPLY] We removed the sentences from the text.

**pg7 ln8 - "suppressed", this should be caveated as there are still significant emissions from African regions, like the Bodele, that are just not transported northwards.**

[REPLY] We thank the reviewer for this comment. Indeed the sentence was generic, and hence not correct. We specify it accordingly (page 9, line 25):

"During autumn and winter the emission and transport of dust towards Europe is suppressed due to the more effective removal processes and due to the atmospheric dynamics favouring the transport of dust towards the Atlantic (e.g. Israelevich et al., 2002; Schepanski et al., 2009)."

**pg7 ln23 - "Strong topographical heights", unclear meaning - please rephrase**

[REPLY] We deleted this part, as it was unclear. Now the sentence is (page 10, line 8):

"The activated dust sources are located in the broad "dust belt" and are usually associated with topographical lows in the arid regions and with the intermountain basins (Prospero et al., 2002)."

**pg8 ln3-5 - It is not clear what this tells us (high DOD, high sdev). Please explain what this indicates.**

[REPLY] Standard deviation is an indication of the variability of the dataset. We deleted this part, as it was unclear.

**pg8 ln3 - "In general,"**

[REPLY] Done.

**pg8 ln31 - "situation", please be more specific.**

[REPLY] We changed the sentence 9Page 11, line 27):

"During OND the horizontal pattern is similar to JJA however with much lower heights (Figs. 3g, h)."

**pg9 ln11 - "England" should be "Ireland"**

[REPLY] It is corrected.

**pg9 ln15 - "England" should be "British Isles"**

[REPLY] It is corrected.

**pg10 ln2 - "higher" than what?**

[REPLY] We rephrased that paragraph. Now the new sentence is (page 13, line 12):

"The values of Clim-DE are higher (>45 $Mm^{-1}$) over Africa during winter and spring, in relation with the ones observed during the other two seasons (<45 $Mm^{-1}$) and reach high altitudes (5-6 $km$ a.s.l.) during spring and summer."

**pg10 ln18 - no new paragraph and replace "Nevertheless" with "However"**

[REPLY] Done.

**pg10 ln23 - delete the first sentence**

[REPLY] We deleted it.

**pg10 ln24 - "Number of Exceedances", exceedances of what? Perhaps "Number of occurences" or "Number of observations" makes more sense?**

[REPLY] We changed the "Number of Exceedances (NoE)" into "Number of dust observations (dO)" according to the suggestion of the reviewer.

**pg11 ln3 - "move" should be "moves"**

[REPLY] It is changed.

**pg11 ln11 - "mean" should be "means"**

[REPLY] It is changed.

**pg11 ln11-20 - this section is out of place as the following paragraph returns to Fig.4. Also, sentences in the paragraph somewhat contradict each other. If the PDR is a means of estimating age, but "cannot be considered as a possible age index". If the latter is true, why is this useful? The paragraph needs moving and restructuring, or removing (which would also mean removing the figure) unless the PDR provides some insight.**

[REPLY] In the figure, were we presented the particle depolarization ratio for the cases used for the production of Cond-DE, it was evident that the depolarization is higher for air masses closer to the desert while it decreases as the air-masses travel towards Europe. This is due to the mixing of dust with other aerosol particles, which takes place after some days of transport. However, the depolarization ratio cannot be considered as a possible age index of the pure-dust particles, since it only provides the mixing of dust with other particles (Tesche et al., 2009). It was used here as an age estimator only because the Sahara desert is away from Europe and the mixing of transported dust with anthropogenic particles occurs as soon as the plumes mix with anthropogenic particles over the European Continent.

Because the revised manuscript, after the suggestion of the reviewers, has two new figures and two new tables, which highlight in our opinion very interesting and informative aspects of the product, we decided to delete the depolarization part (and plot) from the paper, so as to help the reader concentrate on the other parts of this work.

**pg11 ln22 - I think panel "l" should be panel "i"**

[REPLY] It is corrected.

**pg11 ln25 - "plums" should be "plumes"**

[REPLY] It is corrected throuout the manuscript.

**pg11 ln26 - give the latitude range of the mountainous regions**

[REPLY] We elaborated the sentence by including the range of the mountainous regions (page 14, line 26):

"The trapping of Saharan dust from the mountainous ridges of Europe (located between 40°N – 50 °N, e.g. the Alps 45°N-48°N) is also evident by the Con-DE cross-sections(e.g. Fig. 5i, m).

**pg11 ln29 - "dust in" should be "dust at"**

[REPLY] It is changed

**pg11 ln30 - can you be more specific why the deposition is stronger during that season - is it primarily wet or dry deposition?**

[REPLY] We changed the sentence as (page 14, line 29):

"Dry deposition of dust at these areas result also in the formation of "brown snow" and albedo reduction, with profound climatological implications (e.g., Fujita, 2007; Shahgedanova et al., 2013). This phenomenon is more intense during JFM period due to the advection of dust at lower heights."

**pg11 ln31-34 - why is there a sudden drop off in extinction at 40N during the AMJ season? Please explain this.**

[REPLY] We addressed this issue by adding the sentence (figure 14, line 32):

"The transport of dust during AMJ is mostly due to the eastward propagation of N.Africa – Mediterranean low pressure systems (Sharav cyclones). Dust is embedded in the cyclonic circulation and the penetration to latitudes higher than 40°N is limited."

**pg12 ln1 - I think panel "i" should be panel "I"**

[REPLY] It is corrected.

**pg12 ln19-24 - the studies referenced consider longer time periods and/or different geographical regions, please alter to so that the discussion relates better to the region and period you are considering.**

[REPLY] As mentioned already in the similar major comment, the "Interannual variability of dust" section is modified by adding the following text (page 15, line 29):

"In comparison with studies relevant to the time period considered in this work, the DOD decrease of 0.001 $yr^{-1}$ over the northern coast of Africa is in agreement with Floutsi et al. (2016), who based on 12 years of MODIS-Aqua observations (2002-2014) reported an average decrease of 0.003 $yr^{-1}$ for the coarse mode fraction of AOD over the broader Mediterranean Sea. Furthermore, over the same domain the decreasing trend of DOD coincides with the decrease of Saharan desert dust episodes as reported by Gkikas et al. (2013). Regarding the AERONET stations over the domain of northern Africa and Europe, Yoon et al. (2012) reported on the trends of AOD at 440 nm along with the corresponding Ångström Exponents (440 and

870nm). The documented negative trends over the AERONET stations of Avignon (France), Dakar (Senegal) and Ispra (Italy) are in agreement with the negative DOD reported here, although with discrepancies in the magnitude, while trend disagreements are observed over the AERONET station of Banizoumbou (Niger). The decreasing trends of DOD observed over the domain northern of Africa and Europe coincide with the generally documented downward AOD trends reported based on several satellite observations of MODIS/Aqua, MODIS/Terra, MISR and SeaWiFS (Pozzer et al., 2015; de Meij et al., 2012; Hsu et al., 2012; Georgoulias et al. 2016b). More particular, in the most recent study of Georgoulias et al. (2016b), using MODIS/Terra and MODIS/Agua observations, they reported negative statistically significant trends over Algeria, Egypt and the Mediterranean and positive trends over Middle East. Overall, for the Mediterranean they reported an AOD trend of -0.0008 $yr^{-1}$ for the MODIS/Terra observations (2000 – 2015) and -0.0020 $yr^{-1}$ for the MODIS/Aqua observations (2002 – 2015), with the trends being statistical significant at the 95% confidence level in both cases."

**pg12 ln26 - replace LR with lidar ratio**

[REPLY] It is replaced.

**pg24 - The EARLINET reference is repeated multiple times.**

[REPLY] The different EARLINET references refer in different EARLINET publications / products. In particular: (1) EARLINET all observations (2000–2010), 2014a, (2) EARLINET climatology (2000–2010), 2014b, (3) EARLINET correlative observations for CALIPSO (2006–2010), 2014c, (4) EARLINET observations related to volcanic eruptions (2000–2010), 2014d, (5) EARLINET observations related to Saharan Dust events (2000–2010), 2014d.

**Figure 2(b,d,f,h) - Why is the color bar different for the CoM panels relative to the Top Height panels when they are both showing altitude? Consider using the same color to avoid confusion**

[REPLY] We used different range for the altitude in the two plots, because the CoM variation of the area is not nicely depicted when using the same color bar with Top Height. We understand that this might bring confusion to the readers, so we changed the figures using the same color bar and ranges.

**Figure 2 - In titles, "TOP" should be "Top"**

[REPLY] It is changed.

**Figures 3,4,5 - longitude and latitude labels are too small on the domain panels**

[REPLY] The labels size is increased and it is more visible in the new version of the manuscript.

---

## Author Response (AR2)

**The manuscript is improved and most of the comments from the previous review have been addressed. The manuscript is ready for publication after some minor corrections:**

[REPLY] We thank the reviewer who, for a second time, provide us useful comments. Replies to the general and specific comments follow below.

**Page 9**

**Line 8: It would be useful if you plot the boundaries of the regions considered in a map, either in Figure 1 or as an additional figure (maybe as supplementary material?).**

[REPLY] We added a new Figure 2 and changed the manuscript accordingly:

Page 9, line 8: "In order to provide a more quantitative representation of the dataset, the domain is aggregated in six areas over the study region. The main results and statistical parameters are provided in Table 2, and a map with the domains is shown in Fig.2."

**Line 11: Remove "(with 100% as unity)"**

[REPLY] It is removed.

**Page 10**

**Line 14: Because of the color scale, it looks like there is no dust influence above Central and Northern Europe and this sentence might be confusing. Even though values are included in Table 2, you should consider add some text here including information on the values to avoid confusion and/or the use of a logarithmic color scale in the figure.**

[REPLY] We revised this part including information on the mean dust values over Central and Northern Europe accordingly:

Page 10, line 13: "Dust is also present over central and northern Europe with mean DOD up to 0.033 ± 0.062 and occurrence percentages up to 61 % (Fig. 1c; Table 2), revealing that dust particles can be transported far away from their sources under favourable meteorological conditions."

**Line 32: Indicate here if the discrepancies are within the combined uncertainty**

[REPLY] WE changed this part accordingly:

Page 10 line 29: "MODIS provides the AOD for all natural and anthropogenic aerosol types. As a result the MODIS average value for the whole period and domain (0.267) is 281% bigger than our product (0.095 ± 0.04). It is noted though that the values between the two satellite products are very similar over the Sahara desert. On the contrary, the corresponding average dust optical depth values of MACC (0.100) and RegCM4 simulations (0.104) consider only dust and are in better agreement with our product, with lower values by 5% and 8.6%, respectively. The 95% confidence interval of the mean for MACC is between 0.092 and 0.108, and for

RegCM4 is between 0.099 and 0.108. Considering these ranges, the discrepancies between CALIPSO dust product and the two models are within the combined uncertainty."

**Page 11**

**Line 16: Include the definition of a.s.e. the first time is used in the text instead of here**

[REPLY] In is corrected.

**Page 12**

**Line 2: Can you add some sentences here explaining how this correlate with the DOD in figure 1?**

[REPLY] This section focuses on the height of the dust layers and how this is consistent with studies employing ground-based lidars. DOD has been discussed elsewhere and in the previous section.

**Line 15: Replace "Spain" by "Iberian Peninsula"**

[REPLY] It is replased.

**Line 28: Add some sentences in this paragraph explaining how the uncertainties influence your results, especially when discussing values as low as 5 Mm-1.**

[REPLY] We inserted a line that resumes the main point for the uncertainties, as these are presented in the dedicated section. The manuscript is revised accordingly:

Page 12 line 31: "At higher latitudes, the CALIPSO dust extinction is drastically reduced but still observed at 1-2 $km$ a.s.l., with mean Clim-DE values of 5 $Mm^{-1}$. As discussed in detail in Section 2.4, the uncertainty of the dust extinction values close to the surface and at high latitudes is < 54%, with the higher uncertainty in this region mainly originate from the selection of the $\delta_{nd}$ value during the dust separation step. Moreover, the standard deviation, coming from the natural variability of the dust events is an order of magnitude higher than the mean values (Table 3)."

**Page 13**

**Line 16: Consider starting a new paragraph here. Additionally, a reference to Table 3 is missing.**

[REPLY] We deleted the rest of the paragraph here, based on the following comment of the reviewer. Additionally, we added a reference to Table 3 in the beginning of the section:

Page 12 line 8: "To further illustrate the vertical dynamics of dust reaching Europe, the area of study between 20° W and 30° E is separated into five longitudinal zones of 10°, covering latitudes from 20° to 60° N, and the results are presented as latitude-height cross-section plots in Fig.5, with the respective statistics in Table 3."

**Lines 16-22: Consider rewriting this part. As it is now, it's just a listing of the values in Table 3 with little or no discussion.**

[REPLY] Thank you for this comment, it is true that this paragraph had been added after a comment we received from another reviewer. We deleted this paragraph completely.

**Line 22: It is still not clear what the point of including the dust mass concentration values in the study is. Consider removing them from the manuscript.**

[REPLY] We added a comment that explains why we insist on giving the concentration estimations: The main reason is that this information is quite valuable in modelling studies. Page 13 line 23: "The above results are representative of the spatial distribution of dust load as this is approximated by the aerosol extinction coefficient. In order to provide the dust load in units that are more relevant for modelling studies, we estimate here the dust mass concentration."

**Line 28: Start new paragraph here.**

[REPLY] We started a new paragraph.

**Page 16**

**Lines 14-17: Consider moving this information to earlier in the section or rewriting.**

[REPLY] We moved this information earlier in the section. The section now begins as:

Page 15, line 16: "In this section we present the CALIPSO derived monthly mean DOD values, for the total-column and for five individual layers (0.18–0.5, 0.5–1, 1–2, 2–4, 4–8 $km$), in order to study their inter-annual variability during the 9 year period between 2007 and 2015. The selected layers are representative for both near surface and long-range transported dust plumes. The data are aggregated on a 10° x 10° cell over the study region. Using a first-order autoregressive linear regression model on the de-seasonalized monthly DOD values (108 in total) as described in Zanis et al. (2006), temporal trends of DOD were calculated. We note that nine years are considered a small period for a robust trend calculation and it would be interesting to extend this analysis with future measurements. Figure 7 shows the geographical distribution of de-seasonalized trends ($year^{-1}$) for the columnar DOD (a), for the five individual layers (b-f)..."

**Figure 7: Please, change the scale in Figure 7. It's difficult to obtain any information from the plots in the first two rows with the current scale.**

[REPLY] Reply: we changed the scale in this figure (now figure 8). In comparison with the max value of 0.7 (before), the new max value of the first two rows is 0.1.

[Figure]

**Revise the manuscript to check for misspelling and typos:**

**Page 2, line 9: "andupon"**

**Page 10, line 30: Replace "thought" by "though"**

**Page 11, line 18: "kmand"**

**Page 12, Line 16: "kmheight"**

**Page 12, line 17: "kma.s.l."**

[REPLY] We revised the manuscript for misspelling and typos.